# Selective integration of diverse taste inputs within a single taste modality

**Julia U Deere[1†], Arvin A Sarkissian[2], Meifeng Yang[3], Hannah A Uttley[1], Nicole Martinez Santana[1], Lam Nguyen[3], Kaushiki Ravi[3], Anita V Devineni[1,2,3]***

[1]Zuckerman Mind Brain Behavior Institute, Columbia University, New York, United States; [2]Neuroscience Graduate Program, Emory University, Atlanta, United States; [3]Department of Biology, Emory University, Atlanta, United States

**Abstract** A fundamental question in sensory processing is how different channels of sensory input are processed to regulate behavior. Different input channels may converge onto common downstream pathways to drive the same behaviors, or they may activate separate pathways to regulate distinct behaviors. We investigated this question in the *Drosophila* bitter taste system, which contains diverse bitter-sensing cells residing in different taste organs. First, we optogenetically activated subsets of bitter neurons within each organ. These subsets elicited broad and highly overlapping behavioral effects, suggesting that they converge onto common downstream pathways, but we also observed behavioral differences that argue for biased convergence. Consistent with these results, transsynaptic tracing revealed that bitter neurons in different organs connect to overlapping downstream pathways with biased connectivity. We investigated taste processing in one type of downstream bitter neuron that projects to the higher brain. These neurons integrate input from multiple organs and regulate specific taste-related behaviors. We then traced downstream circuits, providing the first glimpse into taste processing in the higher brain. Together, these results reveal that different bitter inputs are selectively integrated early in the circuit, enabling the pooling of information, while the circuit then diverges into multiple pathways that may have different roles.

*For correspondence:
anita.devineni@emory.edu

Present address: †Rockefeller University, New York, United States

## Editor's evaluation

This important manuscript addresses the complexity of processing and representation within bitter taste perception using the *Drosophila* model. The authors provide convincing experimental support for distinct anatomical pathways that process bitter tastes and converge on joint downstream neurons to elicit avoidance responses. The combination of behavioral assays, in vivo physiology, optogenetic manipulation, and connectomics leads the authors to a compelling model of bitter taste processing.

## Introduction

Within a sensory system, sensory neurons are often specialized to detect different types of stimuli. For instance, different mechanoreceptor neurons detect different types of touch (*Abraira and Ginty, 2013*) and different types of cones detect distinct wavelengths of light (*Rushton, 1972*). Different sensory neurons may also detect stimuli in distinct locations, such as touch receptors in different parts of the body or photoreceptors in different parts of the retina. A fundamental question thus arises in each sensory system (*Estebanez et al., 2018*; *Rompani et al., 2017*; *King et al., 2018*; *Haverkamp et al., 2018*; *Bates et al., 2020*; *Jin et al., 2021*): when are distinct channels of sensory input merged and when are they kept separate in order to guide behavior? If they are merged, how and where does this integration occur?

**Figure 1.** Models for bitter taste processing and *Gal4* lines to target bitter neuron subsets. (**A**) Schematic depicting three major taste organs in the fly. (**B**) Models for how different subsets of bitter-sensing neurons could be processed in the brain. (**C–D**) Expression patterns of *Gal4* lines used to target subsets of bitter-sensing neurons. (**C**) Expression of each *Gal4* line in the foreleg, labellum, and pharynx. Endogenous expression of the TdT marker was imaged in flies carrying each *Gal4* along with *UAS-Chrim-TdT*. Autofluorescence that does not represent labeled cells is visible in some pictures, such as at leg joints. (**D**) Table summarizing each expression pattern, based on previous studies cited in the text. Numbers refer to the number of cells per side (left or right) that each line is expressed in relative to the total number of bitter-sensing cells in that organ (leg refers to all three legs). Source data is provided for the table in **D**.

The online version of this article includes the following source data for figure 1:

**Source data 1.** Expression pattern of *Gal4* lines used to target bitter neuron subsets.

Like other sensory systems, the gustatory system contains diverse types of sensory cells. Most studies classify taste cells based on the taste modality that they detect, such as sweet or bitter, which are often studied as homogeneous populations (*Liman et al., 2014*). However, each taste modality contains a repertoire of functionally diverse sensory cells. This diversity has been well characterized in *Drosophila melanogaster*. Flies contain sweet- and bitter-sensing cells in multiple organs, including external organs such as the legs and labellum (the distal segment of the proboscis, the fly's feeding organ) and internal organs such as the pharynx (*Scott, 2018*; *Figure 1A*). Sweet- and bitter-sensing neurons within a single organ also exhibit functional diversity (*Scott, 2018*; *Weiss et al., 2011*; *Ling et al., 2014*; *Fujii et al., 2015*; *Chen and Dahanukar, 2017*). For example, the labellum contains four classes of bitter-sensing neurons (S-a, S-b, I-a, and I-b), defined by the gustatory receptors(Grs) that they express, and these classes show different response profiles to bitter tastants (*Weiss et al., 2011*). Flies also contain taste cells to detect other modalities, such as salt, sour (acid), and fat, but their functional diversity is not as well established (*Jaeger et al., 2018*; *Ahn et al., 2017*; *Tauber et al., 2017*; *Mi et al., 2021*; *Ganguly et al., 2021*).

Thus, a taste modality such as 'sweet' or 'bitter' does not represent a single pathway, but instead comprises multiple channels of sensory input within and across organs. How are these diverse channels

of taste input translated into behavioral output? One possibility is that these input channels converge onto common downstream pathways to regulate the same set of behaviors, representing an 'integrative model' (*Figure 1B*, left). Alternatively, different input channels may be processed separately to regulate distinct behaviors, a 'segregated processing model' (*Figure 1B*, right). This would allow the fly to execute specific behavioral responses depending on the compound that is detected or the organ that detects it.

Previous work, focusing mainly on appetitive tastes, suggests that taste neurons in different organs have different behavioral roles (*Thoma et al., 2016*; *Murata et al., 2017*; *Chen et al., 2021*; *Chen et al., 2022*). Appetitive tastes such as sugar elicit a sequence of behaviors, with specific organs regulating each step (*Dethier, 1976*; *Scott, 2018*; *Mahishi and Huetteroth, 2019*). Flies first detect the presence of sugar with their legs. Depending on the sugar concentration and the fly's hunger state, taste detection by the leg may cause the fly to stop walking and extend its proboscis to contact the food. Labellar taste stimulation may then trigger the initiation of food consumption. Food consumption leads to stimulation of pharyngeal taste neurons, which regulate the duration of food ingestion (*LeDue et al., 2015*; *Yapici et al., 2016*; *Joseph et al., 2017*) and elicit local search behavior to find more food (*Murata et al., 2017*). Thus, sugar-sensing inputs from different organs appear to be processed separately to regulate different aspects of behavior.

However, it is not clear whether the same principle applies to aversive tastes such as bitter. Some studies suggest that different types of bitter neurons have different behavioral roles (*Joseph and Heberlein, 2012*; *Chen et al., 2019*), but this has not been widely examined. The anatomical organization of sensory projections also suggests the possibility of segregated processing: bitter-sensing neurons in different organs project axons to different regions of the subesophageal zone (SEZ), the primary taste area of the brain, suggesting potential connections to distinct downstream pathways (*Wang et al., 2004*; *Kwon et al., 2014*). Finally, the observation that bitter-sensing neurons in different organs show different response dynamics suggests that these neurons may have distinct functional roles (*Devineni et al., 2021*).

In this study, we leverage a wide array of genetic tools and behavioral assays to systematically address this longstanding question: are different channels of bitter input integrated or processed separately to regulate behavior? First, we optogenetically activated subsets of bitter neurons in different organs. Bitter neuron subsets elicited a broad and highly overlapping set of behaviors, suggesting that these inputs converge onto common downstream pathways, but we also observed differences that argue for biased convergence. Next, we used transsynaptic tracing to determine whether different types of bitter neurons connect to common or distinct pathways. Consistent with our behavioral results, bitter neurons in different organs connect to overlapping second-order pathways with biased connectivity. Finally, we studied a novel type of putative second-order neuron that projects to the higher brain. We show that it integrates bitter input from multiple organs, and we characterized its response properties, behavioral role, and downstream connectivity. Together, these results suggest that bitter inputs from different organs are selectively integrated at the first synapse and drive largely overlapping behaviors, contrasting with the prevailing view that different taste organs have different functions. Our studies also provide insight into the functional role of downstream bitter neurons and the organization of taste pathways in the higher brain.

## Results

### Subsets of bitter neurons in different organs act in parallel to regulate feeding

The goal of this study was to uncover the logic and circuit architecture underlying how different channels of bitter input are processed in the brain. Different bitter inputs, including input from different taste organs or neuronal classes, could be integrated in the brain to regulate the same behaviors (*Figure 1B*, left) or processed separately to regulate distinct behaviors (*Figure 1B*, right). We first aimed to distinguish between these models by optogenetically activating subtypes of bitter neurons within specific organs and examining their effect on a wide range of behaviors. We chose to perform activation experiments rather than neuronal silencing because we wanted to determine which behaviors each set of neurons is capable of driving, which implies connectivity to downstream behavioral circuits. Neuronal silencing may not reveal the same effects due to redundant functions of bitter

neurons, especially because we are only targeting a subset of bitter neurons within each organ (see below). Moreover, optogenetic activation allows precise control over the timing and duration of taste stimulation. As a control, for each experiment we activated the entire population of bitter-sensing neurons and ensured that the resulting behavioral effects are consistent with known effects of bitter taste.

We used previously characterized *Gal4* lines (*Weiss et al., 2011*; *Ling et al., 2014*; *Kwon et al., 2014*; *Chen and Dahanukar, 2017*) to target all bitter-sensing neurons across the body (*Gr33a-Gal4*) as well as four subsets of bitter neurons within specific organs (*Figure 1C–D*): (1) bitter neurons in the leg (*Gr58c-Gal4*); (2) bitter neurons in the labellum belonging to classes S-a and I-a (*Gr59c-Gal4*); (3) bitter neurons in the labellum belonging to class S-b (*Gr22f-Gal4*); and (4) a pharyngeal bitter neuron termed V6 (*Gr9a-Gal4*). The expression patterns of these four *Gal4* lines have been carefully validated by previous studies, showing that each line is expressed in only one of the three taste organs under study (leg, labellum, or pharynx) (*Weiss et al., 2011*; *Ling et al., 2014*; *Kwon et al., 2014*; *Chen and Dahanukar, 2017*). We also imaged these organs to confirm the organ-specific expression pattern of each line (*Figure 1C*).

Because these *Gal4* lines label different numbers of neurons (*Figure 1D*) and may also vary in their expression strength, we focused on analyzing whether the activation of each neuronal subset could drive any significant change in each behavior rather than comparing the strength of behavioral effects across lines. We used the light-activated channel Chrimson to drive neuronal activation (*Klapoetke et al., 2014*). Chrimson has been previously used to activate taste sensory neurons (*Klapoetke et al., 2014*; *Musso et al., 2019*; *Moreira et al., 2019*), and bitter-sensing neurons expressing Chrimson are effectively activated by red light (*Devineni et al., 2021*).

We began by examining the effect of bitter neuron activation on feeding responses. We first tested the proboscis extension response (PER), an appetitive response that represents the initiation of feeding and is suppressed by bitter compounds (*Dethier, 1976*; *Scott, 2018*; *Wang et al., 2004*). As expected, activating the entire population of bitter-sensing neurons strongly suppressed PER to sugar (*Figure 2A*). Interestingly, activating any of the four bitter neuron subsets also suppressed PER to a similar extent (*Figure 2A*). Thus, subsets of bitter neurons within any of the three organs – the leg, labellum, or pharynx – can suppress the initiation of feeding.

PER is an immediate response that is tested in immobilized flies. To examine more naturalistic feeding behavior in freely moving flies over a longer time period, we used the optoPAD assay (*Moreira et al., 2019*). The optoPAD uses a capacitance sensor to detect feeding events, which consist of individual 'sips' clustered into feeding bursts and feeding bouts (*Figure 2—figure supplement 1A*; *Itskov et al., 2014*). We used closed-loop light stimulation to optogenetically activate bitter neurons whenever the fly interacted with one food source (100 mM sucrose), while a second identical food source not linked to optogenetic stimulation served as the control (*Figure 2B*). Activating all bitter-sensing neurons or any of the four subsets caused a near-complete suppression of feeding (*Figure 2C–D*). This suppression was evident in many different feeding-related parameters. All four bitter neuron subsets suppressed the total feeding duration, number of feeding bouts, sip number, and sip duration (*Figure 2D*). The number of feeding bursts was suppressed by all subsets except for leg neurons, although they showed the same trend (*Figure 2—figure supplement 1B*). The duration of individual feeding bouts was suppressed by only one subset, the pharyngeal neurons, suggesting that this specific parameter may be primarily regulated by the pharynx (*Figure 2—figure supplement 1B*).

We next asked whether silencing these bitter neuron subsets prevents flies from suppressing feeding in the presence of natural bitter taste. We used the light-gated chloride channel GTACR to optogenetically silence neuronal activity (*Mohammad et al., 2017*). We presented flies with two bitter-containing food sources (50 mM sucrose + 1 mM quinine), and we used closed-loop stimulation to silence bitter neurons whenever the fly interacted with one of the two food sources (*Figure 2E*). As expected, silencing all bitter neurons across the body strongly increased feeding (*Figure 2F–G*). We did not expect that silencing bitter neuron subsets would have a strong effect because each subset alone robustly suppresses feeding (*Figure 2C–D*), suggesting that they act redundantly. Silencing two of the four subsets (neurons in the leg and pharynx) increased feeding in experimental flies, with pharyngeal neurons showing a stronger effect (*Figure 2F*). However, the effects of light stimulation generally did not reach statistical significance when compared with genetic controls (*Figure 2G* and

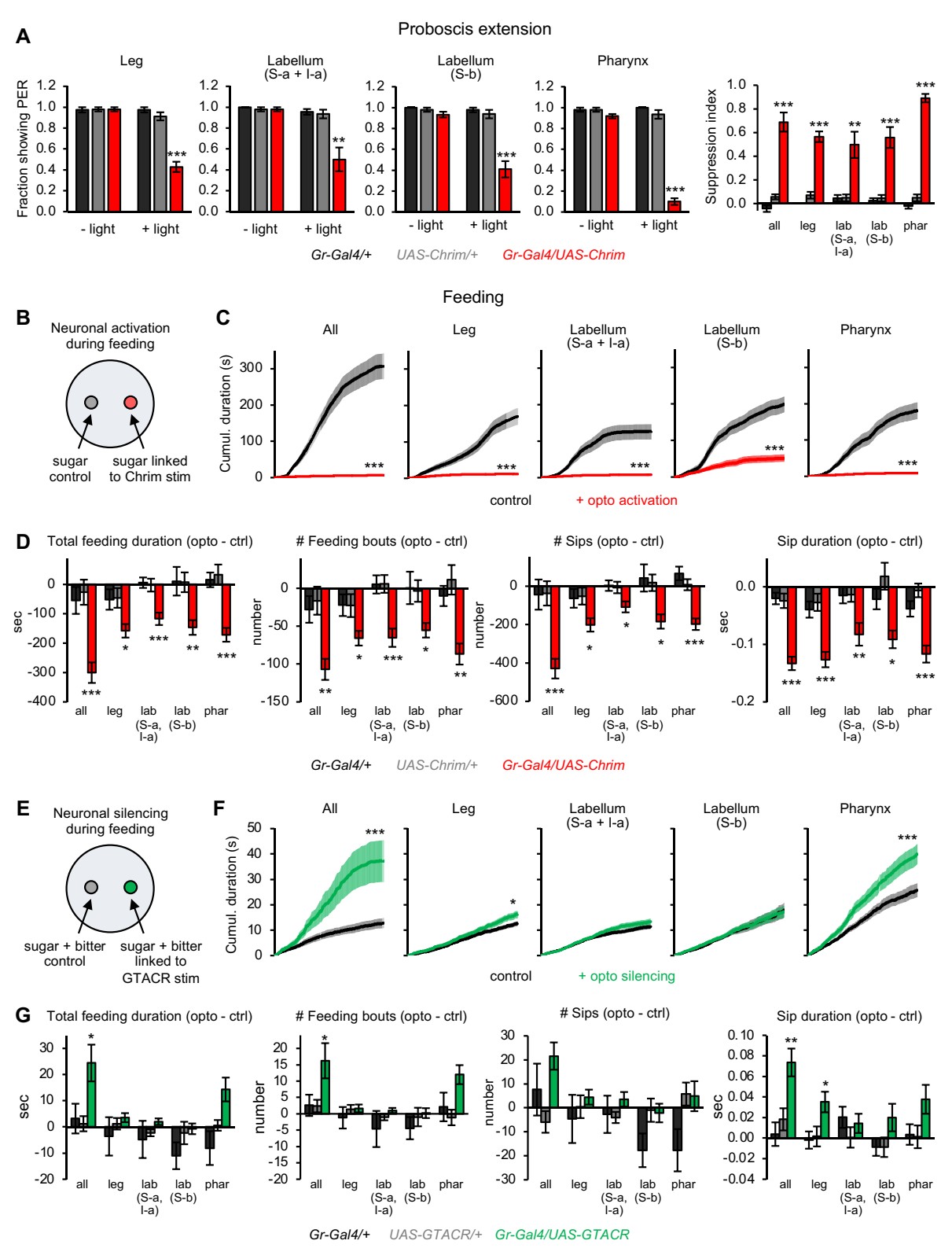

**Figure 2.** Bitter neuron subsets in different organs act in parallel to regulate feeding responses. (**A**) Effect of activating bitter neuron subsets on proboscis extension response (PER) to 100 mM sucrose (n=5–8 sets of flies). Bar graphs for each neuronal subset (left) represent the fraction of flies showing PER with and without light stimulation. Bar graph on the right shows the degree of PER suppression elicited by neuronal activation, quantified as: 1 – (PER with light/PER without light). (**B, E**) Schematic of optoPAD setup used to test how neuronal activation (**B**) or silencing (**E**) affects feeding.

*Figure 2 continued on next page*

*Figure 2 continued*

Food sources contained 100 mM sucrose for activation experiments (**B**) and 50 mM sucrose +1 mM quinine for silencing experiments (**E**). (**C, D, F, G**) Effects of bitter neuron activation (**C–D**; n=40–48 flies) or silencing (**F–G**; n=40–62 flies) on feeding for 1 hr. (**C, F**) Cumulative feeding duration on the control and opto stim food sources for experimental flies. Values for the last time point (1 hr) were compared for control versus opto stim using paired t-tests. (**D, G**) Bars represent the difference in each feeding parameter between the control and opto stim food for each genotype. For each feeding measure, experimental flies were compared to both controls, and effects not labeled with an asterisk are not significant. In all panels, *Gal4* lines used were *Gr33a-Gal4* (all bitter neurons), *Gr58c-Gal4* (leg), *Gr59c-Gal4* (labellum, S-a + I-a), *Gr22f-Gal4* (labellum, S-b), and *Gr9a-Gal4* (pharynx). For all figures: *p<0.05, **p<0.01, ***p<0.001, ns = not significant (p>0.05). Unless otherwise specified, two groups were compared using unpaired t-tests and more than two groups were compared using one-way ANOVA followed by Dunnett's test comparing experimental flies to each control.

The online version of this article includes the following figure supplement(s) for figure 2:

**Figure supplement 1.** Additional characterization of the effects of bitter neuron manipulations on feeding.

*Figure 2—figure supplement 1C*). Thus, activating each bitter neuron subset had strong effects on feeding whereas silencing had weaker or null effects. These results suggest that different bitter neuron subsets act in a parallel and partially redundant manner to regulate feeding, implying that these input channels are integrated by downstream feeding circuits.

## Bitter neuron subsets in different organs elicit similar effects on locomotion

Although different bitter neuron subsets elicited similar effects on feeding behavior, they may exert different effects on other taste-related behaviors. We tested their effect on locomotion, which is known to be affected by taste (*Flood et al., 2013*; *Thoma et al., 2016*; *Tadres et al., 2020*). To monitor locomotor behavior, we filmed flies and tracked their movement in a circular arena (*Figure 3A–B*; *Aso et al., 2014b*). To activate bitter neurons, we delivered 5 s light stimulation at three different intensities ('low', 'medium', and 'high', corresponding to 4, 20, and 35 µW/mm$^2$). Activating the entire population of bitter neurons strongly stimulated locomotion at all light intensities. Light onset elicited a transient increase in turning and a more sustained increase in forward velocity and the fraction of flies moving (*Figure 3C–D* and *Figure 3—figure supplement 1A*). When light stimulation was turned off, flies immediately stopped moving, and this locomotor suppression persisted for nearly a minute (*Figure 3C* and *Figure 3—figure supplement 1A–B*). We observed the same behavioral effects with continuous light stimulation (*Figure 3C–D*) and 50 Hz pulsed light (*Figure 3—figure supplement 1C–D*). Together, these results suggest that when a fly senses a bitter stimulus, it displays a sequence of stereotyped behaviors mediating aversion: (1) it first turns in an attempt to orient away from the stimulus, (2) it increases forward velocity in an attempt to run away from the stimulus, and (3) when the stimulus disappears, it freezes because it may perceive its current position to be a safe (bitter-free) location.

We then tested whether these effects could be evoked by subsets of bitter neurons. We hypothesized that locomotor changes may be regulated by bitter neurons in the leg, analogous to the role of tarsal sweet-sensing neurons in mediating locomotor responses to sugar (*Thoma et al., 2016*). Instead, activating any of the four bitter neuron subsets enhanced locomotion for at least one of the three light intensities (*Figure 3E* and *Figure 3—figure supplement 2A*). Three of the four subsets (all except labellar S-b neurons) also elicited an increase in turning at light onset as well as locomotor suppression after light was turned off, although the latter effect was only significant for two lines (*Figure 3E–F* and *Figure 3—figure supplement 2*). These results show that bitter neurons in three different organs can drive locomotor changes typical of aversion, again suggesting that different bitter inputs converge onto common behavioral circuits.

## Bitter neuron subsets in multiple organs can elicit innate and learned aversion

We next tested the effect of bitter neuron activation on innate and learned aversion. Flies avoid residing in areas containing bitter compounds (*Marella et al., 2006*; *Joseph and Heberlein, 2012*). This positional aversion may correlate with feeding aversion, but it can be displayed in the absence of feeding and is mediated by different motor circuits (i.e., motor neurons controlling the legs versus the proboscis). Positional aversion may also be related to locomotor responses (*Figure 3*), as stimulation of speed and turning would increase the likelihood that flies leave a bitter substrate. However, it is not

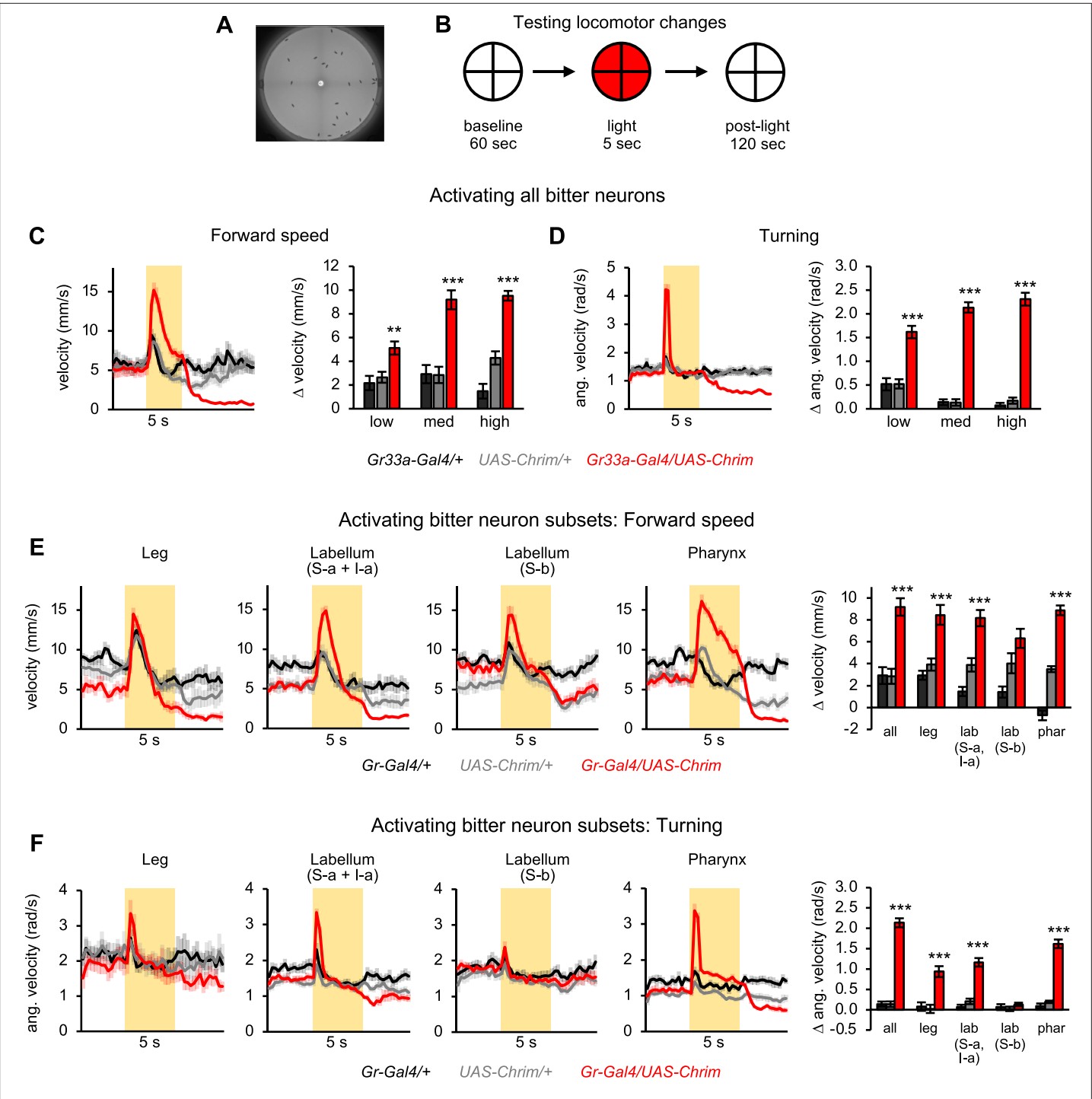

**Figure 3.** Bitter neuron subsets in three different organs elicit similar effects on locomotion. (**A**) Arena used for tracking flies during optogenetic activation. (**B**) Schematic of protocol to test the effect of bitter neuron activation on locomotion. 5 s light stimulation was used at three different light intensities (low, medium, high). (**C–D**) Locomotor effects elicited by activating all bitter-sensing neurons using *Gr33a-Gal4* (n=11–12 trials). Left graphs show forward (**C**) and angular (**D**) velocity over time for medium light intensity (shading indicates light on). Bar graphs quantify the change in these parameters at light onset. (**E–F**) Locomotor effects elicited by activating bitter neuron subsets with medium intensity light (n=10–12 trials). Effects of other light intensities are shown in *Figure 3—figure supplement 2*. Traces show forward (**E**) or angular (**F**) velocity over time (shading indicates light on). Bar graphs quantify the change in these parameters at light onset. *Gal4* lines used were *Gr33a-Gal4* (all bitter neurons), *Gr58c-Gal4* (leg), *Gr59c-Gal4* (labellum, S-a + I-a), *Gr22f-Gal4* (labellum, S-b), and *Gr9a-Gal4* (pharynx). Experimental effects not labeled with an asterisk are not significant.

The online version of this article includes the following figure supplement(s) for figure 3:

*Figure 3 continued on next page*

**Figure supplement 1.** Additional characterization of locomotor effects elicited by activating all bitter neurons.

**Figure supplement 2.** Additional characterization of locomotor effects elicited by subsets of bitter neurons.

obvious that activating each bitter neuron subset should have an identical effect in both behavioral assays, as aversion is measured over a longer timescale and active choice processes may contribute.

To measure innate aversion, we presented light in two opposing quadrants of the arena and quantified the flies' preference for residing in the light quadrants, where they experience bitter neuron activation (*Figure 4A*). Expressing Chrimson in all bitter neurons elicited strong positional aversion at all light intensities (*Figure 4B* and *Figure 4—figure supplement 1A*). The effect was similar for continuous or 50 Hz pulsed light stimulation (*Figure 4—figure supplement 1B*). We then tested the effect of each bitter neuron subset. Activating three of the four bitter neuron subsets elicited innate aversion (*Figure 4C* and *Figure 4—figure supplement 2*). This included neuronal subsets in the leg, labellum, and pharynx, with only labellar S-b neurons failing to elicit an effect.

In addition to driving innate aversion, bitter taste can serve as a negative reinforcement signal to elicit learned aversion to odors (*Das et al., 2014*). We used previously established protocols (*Aso and Rubin, 2016*) to test whether bitter neuron activation could drive learned odor aversion. We delivered one odor (the conditioned stimulus, CS+) while activating bitter neurons, then delivered a different odor (the CS-) without neuronal activation, and finally allowed the flies to choose between quadrants containing the CS+ versus CS- (*Figure 4D*). As expected, pairing odor with the activation of all bitter neurons elicited aversion to the CS+ (*Figure 4E*). Activating two of the four bitter neuron subsets also elicited learned aversion (*Figure 4F*), including neurons in the leg and labellum. Labellar S-b neurons failed to elicit a significant effect, similar to their lack of an effect on innate aversion. Interestingly, pharyngeal neurons did not elicit a learned response despite driving strong innate aversion.

Together, these experiments demonstrate that different subsets of bitter neurons drive broad and largely overlapping behavioral responses (*Figure 5A*). The fact that all neuronal subsets strongly evoked at least one aversive behavior suggests that each set of neurons was effectively activated by Chrimson. Labellar S-b neurons had the sparsest effect and only affected three of the seven behavioral measures analyzed (*Figure 5A*), but we noted that the corresponding *Gal4* line may have weaker expression than the other lines (based on weaker labeling of axonal projections in the brain and the inability to drive consistent *trans*-Tango labeling in the experiments below; data not shown). The other three subsets, residing in three different organs, all affected at least six of the seven behaviors tested, including proboscis extension, feeding, locomotion, and innate aversion (*Figure 5A*). Two of those three subsets (leg and labellar S-a + I-a neurons) also caused aversive learning.

These results generally support an integrative model of taste processing (*Figure 1B*, left) in which bitter inputs from different organs converge onto common downstream pathways to drive a diverse set of aversive behaviors. However, the fact that not all neuronal subsets elicited all behaviors, or preferentially elicited some behaviors over others, suggests that some downstream pathways may receive selective or preferential input from specific bitter-sensing cells. We refer to this model as 'selective integration' (*Figure 5B*), and it lies in between the two models shown in *Figure 1B*.

## Bitter input from different organs is relayed to overlapping downstream regions

Our behavioral results suggest that taste circuits in the brain integrate bitter input from multiple organs. Where does this integration occur? Cross-organ integration could occur early in the taste circuit, potentially at the first synapse, or much later in the circuit. To test whether integration may occur at the first synapse, we used the transsynaptic tracing method *trans*-Tango (*Talay et al., 2017*) to identify neurons that are postsynaptic to bitter sensory cells, termed second-order bitter neurons. We note that we cannot use the recently published synaptic connectome of the fly brain (*Scheffer et al., 2020*) to identify second-order neurons because the connectome does not include the SEZ, the region containing bitter sensory axons. Although some gustatory neurons in the legs arborize in the ventral nerve cord, the axons of tarsal bitter-sensing neurons pass through the nerve cord without making arborizations typical of synaptic connections (*Kwon et al., 2014*). We therefore focused on second-order bitter pathways in the brain.

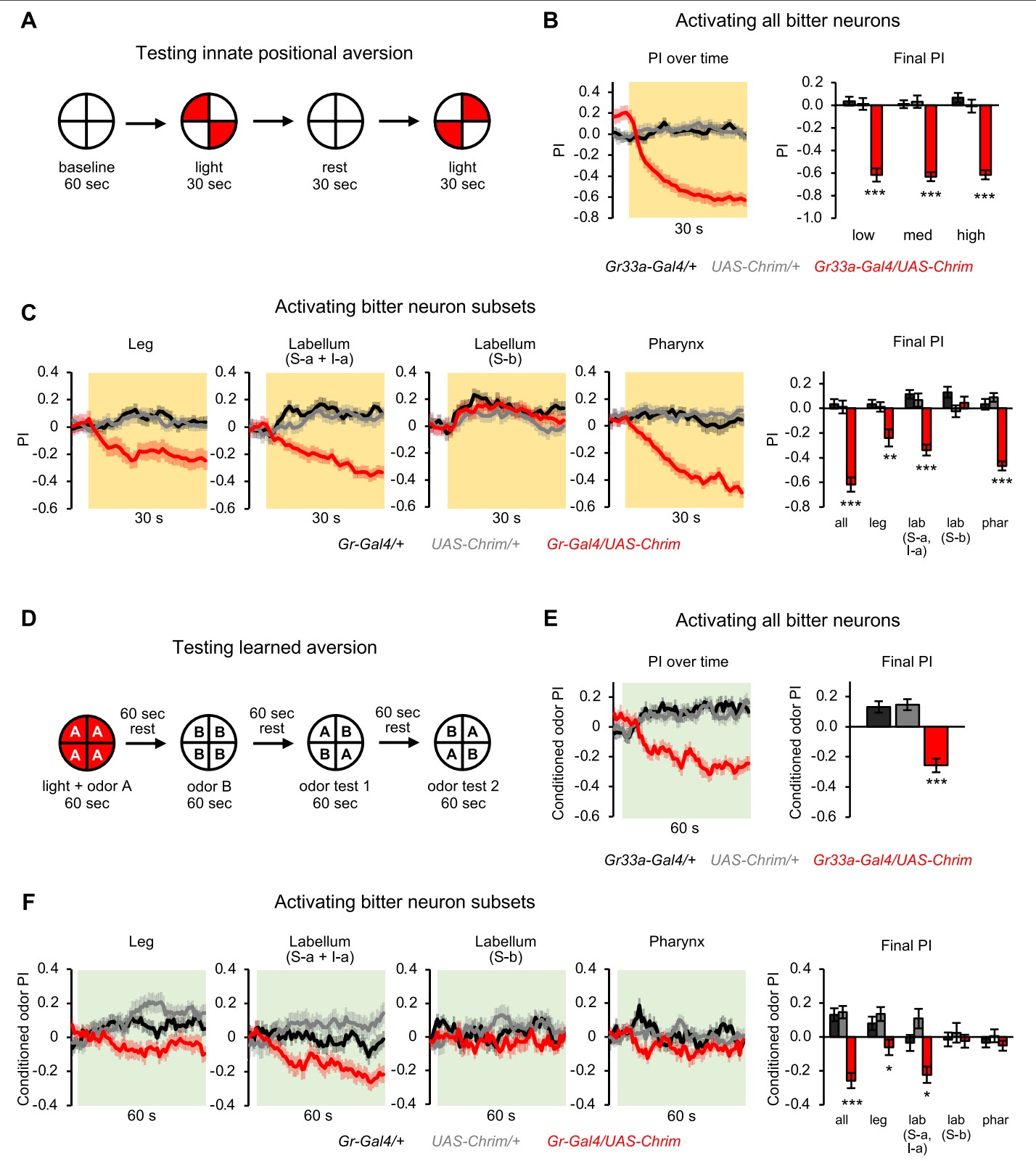

**Figure 4.** Subsets of bitter neurons in multiple organs elicit innate and learned aversion. (**A**) Protocol to test the effect of bitter neuron activation on innate positional aversion. (**B–C**) Effect of activating all bitter-sensing neurons (**B**) or subsets of bitter neurons (**C**) on innate aversion (n=20–24 trials, 10–12 sets of flies). Light preference is quantified by the preference index (PI): (# flies in light quadrants – # flies in non-light quadrants)/total # flies. Negative values indicate aversion. Line graphs show the PI over 30 s of low intensity light stimulation (shading indicates light on), including both of the

*Figure 4 continued on next page*

*Figure 4 continued*

sequential test periods when the lighted quadrants were switched. Flies may appear to show a positive PI before light onset because of the repeated tests: after the first test they continue to avoid the previously illuminated quadrants until the next test. Bar graphs show the final PI (average over last 5 s of light presentation) for each genotype at all light intensities (**B**) or low light intensity (**C**). Effects of bitter neuron subsets at other light intensities are shown in *Figure 4—figure supplement 2*. (**D**) Protocol to test the effect of bitter neuron activation on learned odor aversion. (**E–F**) Effects of activating all bitter-sensing neurons (**E**) or subsets of bitter neurons (**F**) on learned aversion (n=20–24 trials, 10–12 sets of flies). Learned preference was quantified as the PI for the conditioned odor during the test periods, calculated as: (# flies in CS+ quadrants – # flies in CS- quadrants)/total # flies. Negative values indicate aversion. Data from odor tests 1 and 2 are combined. Line graphs show the conditioned odor PI during the odor test periods (shading indicates odors on). Bar graphs show the final PI (average over last 5 s of test) for each genotype. In all panels, experimental effects not labeled with an asterisk are not significant, and *Gal4* lines used were *Gr33a-Gal4* (all bitter neurons), *Gr58c-Gal4* (leg), *Gr59c-Gal4* (labellum, S-a + I-a), *Gr22f-Gal4* (labellum, S-b), and *Gr9a-Gal4* (pharynx). *Figure 4A* has been adapted from Figure 2A from *Aso et al., 2014b* and *Figure 4D* has been adapted from Figure 1D from *Aso and Rubin, 2016*.

The online version of this article includes the following figure supplement(s) for figure 4:

**Figure supplement 1.** Additional characterization of innate aversion elicited by activating all bitter neurons.

**Figure supplement 2.** Effects of bitter neuron subsets on innate positional aversion at other light intensities.

---

We first labeled the entire population of second-order bitter neurons by tracing neurons that are postsynaptic to *Gr33a-Gal4*-expressing cells. We observed many local neurons projecting within the SEZ as well as three major projections to the superior protocerebrum: a lateral, medial, and medio-lateral tract (*Figure 6A* and *Figure 6—figure supplement 1*). This pattern is consistent with second-order bitter neurons labeled in other recent studies (*Chen et al., 2019*; *Snell et al., 2022*). Thus, at the first synapse, the bitter taste circuit diverges into at least three downstream pathways projecting out of the SEZ.

---

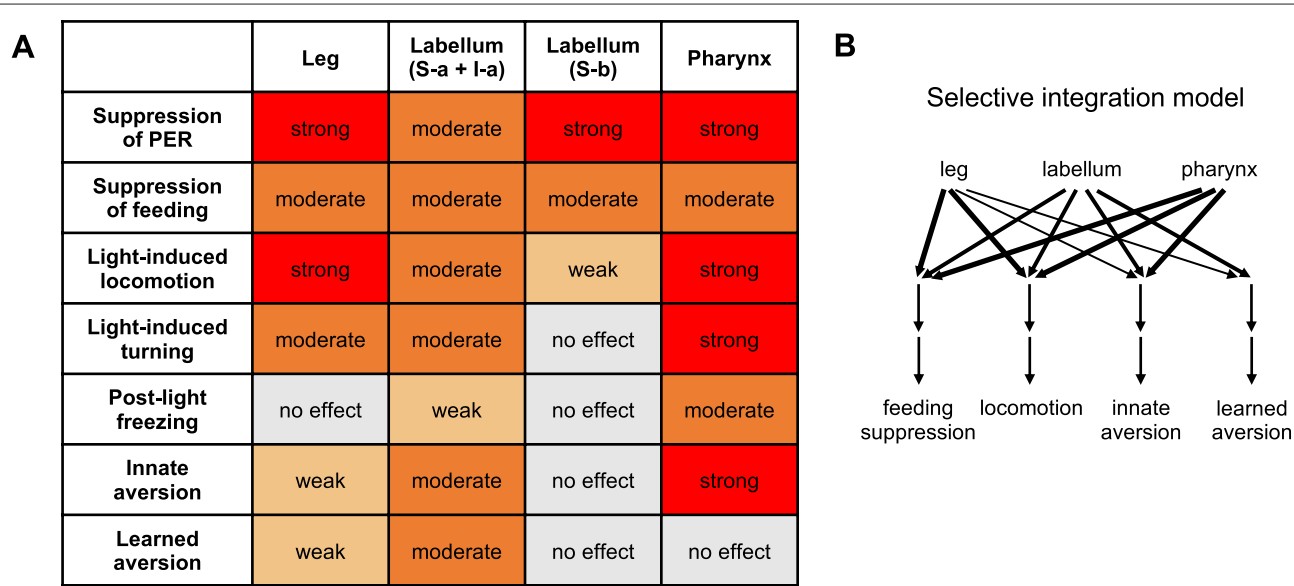

**Figure 5.** Summary of behavioral effects elicited by optogenetic activation of bitter neuron subsets. (**A**) Effects are color-coded by strength, relative to the effect of activating all bitter neurons with *Gr33a-Gal4*. All observed effects went in the same direction. See Materials and methods for details of how effects were quantified. Note that the 'moderate' effect of all subsets on feeding is primarily due to the fact that flies consumed less of the control food compared to activating all bitter neurons (leading to a smaller difference between control and opto stim), but all subsets exerted an almost complete suppression of feeding. (**B**) Model for how different inputs contribute to different types of behavior. Line widths represent the strength of the effect (weak, moderate, or strong). For simplicity, we are only defining four behavioral categories: feeding suppression includes both feeding and proboscis extension response (PER) experiments, and locomotion includes the three measures shown in the table. The strongest effect for each category was used. Results with I-a + S-a neurons (not S-b) were used to interpret the role of the labellum, since positive results are more informative than negative results. Source data is provided for the table in *Figure 5A*.

The online version of this article includes the following source data for figure 5:

**Source data 1.** Summary of behavioral effects elicited by optogenetic activation of bitter neuron subsets.

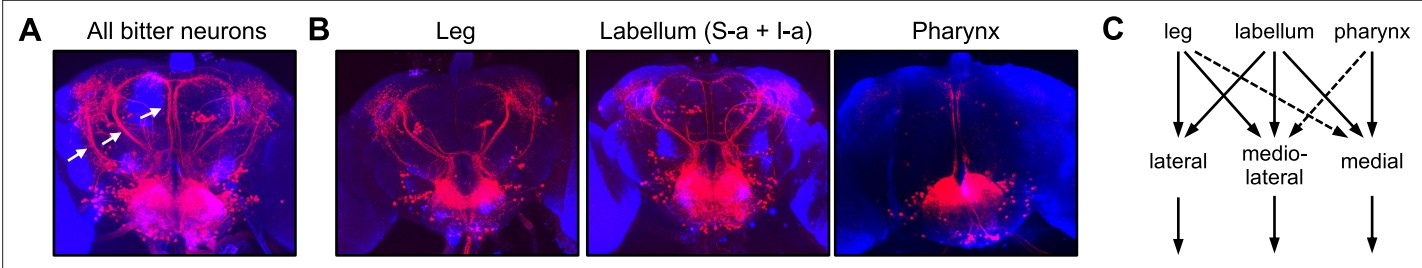

**Figure 6.** Bitter inputs from different organs are relayed to overlapping downstream pathways. (**A**) The entire population of second-order bitter neurons was labeled by *trans*-Tango tracing from *Gr33a-Gal4*-expressing cells. Arrows denote the three major tracts (lateral, mediolateral, and medial) projecting out of the subesophageal zone (SEZ) to the dorsal brain. (**B**) Neurons postsynaptic to specific bitter neuron subsets were labeled with *trans*-Tango. *Gal4* lines used were *Gr58c-Gal4* (leg), *Gr59c-Gal4* (labellar S-a + I-a), and *Gr9a-Gal4* (pharynx). Tracing from labellar S-b cells labeled by *Gr22f-Gal4* yielded very weak *trans*-Tango staining that could not be interpreted, so it is not shown here. (**C**) Summary of inputs from different subsets of bitter neurons onto second-order projection pathways. Dotted lines indicate weak input based on weak or inconsistent *trans*-Tango staining. See ***Figure 6—figure supplement 1*** for additional *trans*-Tango images that include presynaptic *Gal4* expression.

The online version of this article includes the following figure supplement(s) for figure 6:

**Figure supplement 1.** Additional images of *trans*-Tango labeling.

We next used *trans*-Tango to label neurons postsynaptic to specific subsets of bitter sensory cells, using the same *Gal4* lines used to test behavior (***Figure 6B*** and ***Figure 6—figure supplement 1***). Tracing neurons postsynaptic to bitter cells in the leg labeled all three projection tracts, although the medial tract was labeled relatively weakly. Tracing from S-a + I-a neurons in the labellum also labeled all three tracts; the medial tract was labeled more strongly than when tracing from the leg, whereas the lateral tract was labeled more weakly. Finally, tracing neurons postsynaptic to pharyngeal bitter neurons labeled the medial tract almost exclusively, although the mediolateral tract occasionally showed weak labeling. A previous study tracing postsynaptic partners of pharyngeal taste neurons reported similar results, although the mediolateral tract was more clearly labeled in that study (***Chen et al., 2019***).

Together, these results show that: (1) bitter neurons in each organ synapse onto multiple second-order pathways, and (2) each projection tract receives input from multiple organs, although it is not clear whether these inputs converge onto the same individual neurons. Interestingly, bitter inputs show biased connectivity to second-order taste regions, with neurons in different organs preferentially connecting to different projection tracts (***Figure 6C***). This biased connectivity could allow downstream taste pathways to be selectively or preferentially activated by certain organs, as suggested by our behavioral results (***Figure 5B***).

## mlSEZt neurons integrate bitter input from the leg and labellum

In order to study how specific types of second-order bitter neurons encode and integrate bitter information, we require genetic driver lines to label them. By searching the Janelia FlyLight expression database (***Jenett et al., 2012***), we identified multiple *Gal4* lines that appeared to label second-order neurons comprising the mediolateral tract. We refer to these cells as mlSEZt neurons based on the terminology in ***Talay et al., 2017***. We focus the rest of this study on mlSEZt cells, which represent one of the only types of second-order bitter neurons to be functionally studied.

As visible in the first line we identified, *R29F12-Gal4*, mlSEZt neurons have cell bodies just dorsal to the antennal lobes, a V-shaped projection in the SEZ, and a circular projection in the superior lateral protocerebrum (SLP) (***Figure 7A***, top). Colabeling revealed that the axons of bitter-sensing neurons overlap with the medial branch of the mlSEZt projection in the SEZ (***Figure 7A***, bottom). To confirm that mlSEZt neurons receive bitter input, we monitored their responses to taste stimulation using calcium imaging. The projections of mlSEZt neurons in the SEZ and SLP could be clearly identified based on their location and morphology. mlSEZt projections in both the SEZ and SLP responded to bitter stimulation of either the labellum or leg (***Figure 7B–D***), demonstrating that these neurons receive bitter input from multiple organs. Water or sugar applied to the labellum elicited a much weaker response than bitter in the SEZ (~50% of the bitter response; ***Figure 7B***) and no response in the SLP (***Figure 7C***). Water or sugar responses were not observed in either the SEZ or SLP with tarsal

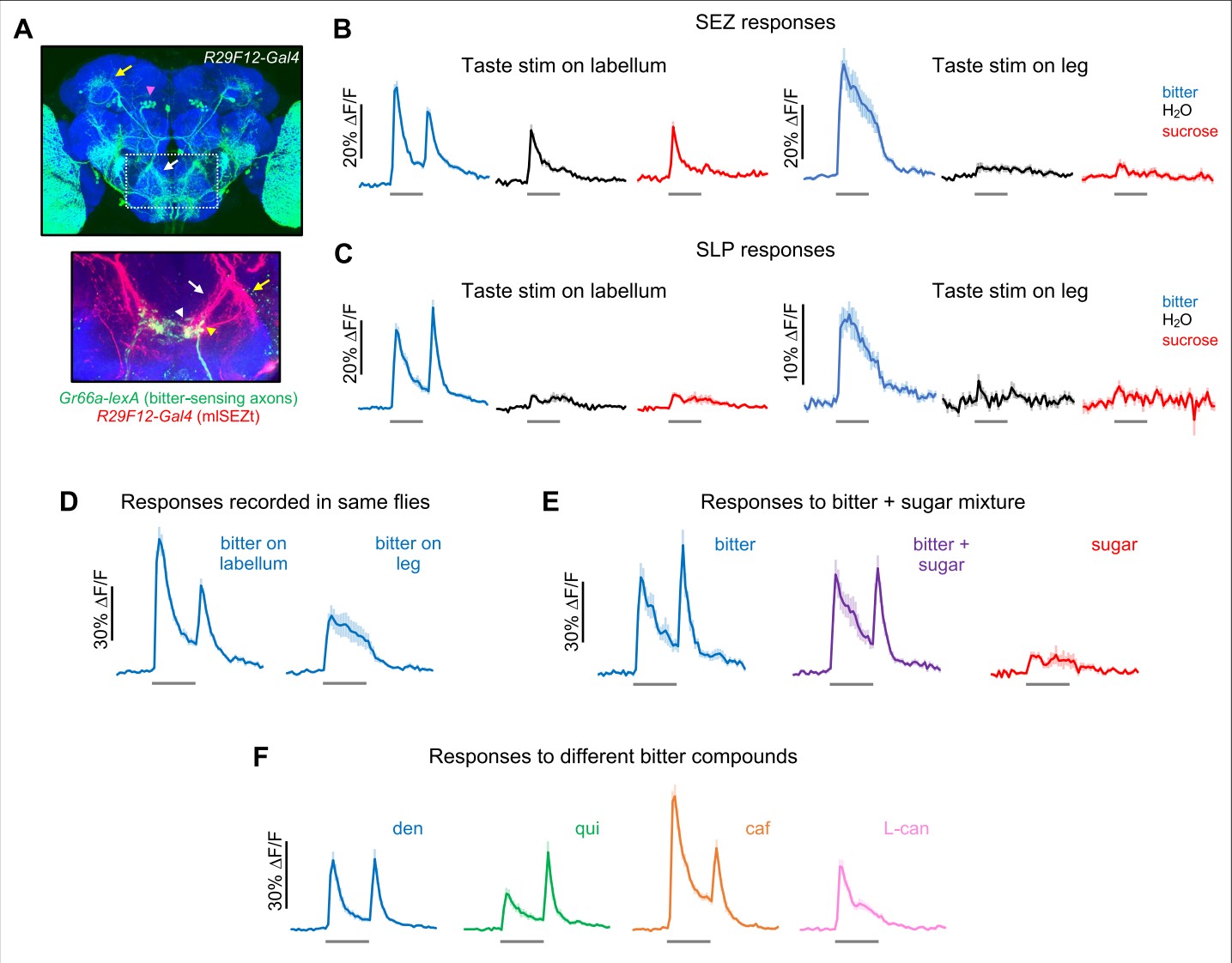

**Figure 7.** Second-order mlSEZt bitter neurons receive input from multiple organs. (**A**) Top: *R29F12-Gal4* expression pattern (maximum intensity projection created from images generated by the Janelia FlyLight Project Team). Brain slices at the far anterior and posterior edges are not included to maximize visibility of mlSEZt cells. Pink arrowhead shows mlSEZt cell bodies; white and yellow arrows show mlSEZt projections in the subesophageal zone (SEZ) and superior lateral protocerebrum (SLP), respectively. Box depicts approximate area of the SEZ shown in the bottom image. Bottom: SEZ image with mlSEZt neurons (*R29F12-Gal4* driving *UAS-TdTomato*, red) colabeled with bitter-sensing neurons (*Gr66a-lexA* driving *lexAop-GCaMP6f*, green). Medial and lateral branches of the mlSEZt projections are denoted by white and yellow arrows, respectively. Based on morphology, most of the visible bitter sensory projections likely arise from the labellum (white arrowhead); bitter axons from the leg have a stick-like projection just lateral to the labellar projections (yellow arrowhead). (**B–F**) GCaMP responses of mlSEZt neurons. *R29F12-Gal4* was used to drive *UAS-GCaMP6f*. For all imaging traces, the gray bar denotes 5 s taste presentation. Unless otherwise specified, the bitter stimulus was 10 mM denatonium and the sugar stimulus was 100 mM sucrose. (**B–C**) GCaMP responses of mlSEZt projections in the SEZ (**B**) or SLP (**C**) with taste stimuli applied to the labellum (left) or foreleg (right). Response magnitudes across these four conditions are not directly comparable because they include different sets of flies. (**B**) n=15–22 trials, 4 flies (labellar stimulation); n=15–19 trials, 4 flies (leg stimulation). (**C**) n=25–34 trials, 7 flies (labellar stimulation); n=12–19 trials, 3–4 flies (leg stimulation). (**D**) Responses of mlSEZt projections in the SEZ to labellar or tarsal bitter stimulation imaged in the same flies (n=11–14 trials, 3 flies). (**E**) Responses of mlSEZt projections in the SLP when sugar, bitter, or a sugar-bitter mixture was applied to the labellum (n=16 trials, 4 flies). (**F**) Responses of mlSEZt projections in the SLP when different bitter compounds were applied to the labellum (n=22–23 trials, 6 flies).

The online version of this article includes the following figure supplement(s) for figure 7:

**Figure supplement 1.** Further characterization of mlSEZt responses.

**Figure supplement 2.** Responses of mlSEZt cells labeled by *R55E01-Gal4* and *R29F12-R55E01 split-Gal4*.

stimulation (*Figure 7B–C*). The addition of sugar did not modulate the response to labellar bitter stimulation (*Figure 7E*), suggesting that mlSEZt neurons do not receive inhibition from the sugar pathway.

mlSEZt neurons showed different response dynamics when bitter was applied to the labellum versus the leg, as previously reported for bitter sensory neurons in these organs (*Devineni et al., 2021*). Bitter stimulation of the labellum elicited transient responses at bitter onset (ON response) and offset (OFF response), whereas stimulation of the leg elicited a more sustained ON response with no OFF response (*Figure 7B–D*). When comparing responses in the same flies (*Figure 7D*), labellar stimulation elicited stronger ON responses than tarsal stimulation (~70% versus ~30% ΔF/F).

Tarsal and labellar bitter stimulation also produced different spatial patterns of activation in the SEZ (*Figure 7—figure supplement 1A–B*). Responses to tarsal stimulation were localized to a small region located at the dorsolateral edge of the larger region activated by labellar stimulation. This is consistent with our colabeling experiment (*Figure 7A*, bottom) and the known projection patterns of tarsal and labellar bitter-sensing neurons (*Wang et al., 2004*; *Kwon et al., 2014*). Thus, bitter inputs from different organs likely activate different dendrites on mlSEZt neurons.

In the SLP, the spatial pattern of mlSEZt activation was similar when bitter was applied to either the leg or labellum (*Figure 7—figure supplement 1C*). We observed individual puncta, likely representing axon terminals of single mlSEZt cells, that were activated by bitter stimulation of either organ. These results suggest that individual mlSEZt cells receive convergent input from the labellum and leg. mlSEZt cell bodies, which are located far from either the SEZ or SLP projections (*Figure 7A*), did not respond to bitter taste (data not shown), preventing us from unequivocally imaging responses in single mlSEZt cells.

Finally, we tested whether bitter responses in mlSEZt neurons depend on the identity of the bitter compound and the recent history of bitter stimulation, as shown for labellar bitter sensory neurons (*Devineni et al., 2021*). Indeed, mlSEZt neurons showed compound-dependent dynamics that resembled those of sensory neurons. Relative to the strength of the ON response, denatonium and quinine elicited strong OFF responses, caffeine elicited a slightly weaker OFF response, and L-canavanine elicited no OFF response at all (*Figure 7F*). mlSEZt responses also showed an experience-dependent effect resembling that of bitter-sensing neurons: with repeated bitter stimulation, the bitter ON response habituated much more strongly than the OFF response (*Figure 7—figure supplement 1D*).

Together, these results strongly suggest that the mlSEZt neurons labeled by *R29F12-Gal4* are second-order bitter neurons: they have an identical morphology to second-order bitter neurons labeled by *trans*-Tango, their projections overlap with the axon terminals of bitter-sensing neurons, and they show clear bitter responses. We will therefore refer to mlSEZt neurons as second-order bitter neurons, although it is formally possible that they are downstream bitter neurons that are not directly connected to bitter-sensing cells. The taste selectivity and response dynamics of mlSEZt neurons closely resemble those of bitter sensory neurons. The major transformation we observe is that mlSEZt neurons receive convergent bitter input from at least two different organs, the leg and labellum. This convergence may represent a substrate by which bitter neurons in different organs elicit common behavioral effects (*Figure 5*).

## mlSEZt neurons regulate a subset of taste-related behaviors

Given the existence of several second-order bitter pathways (*Figure 6*), we wondered how individual types of second-order neurons like mlSEZt contribute to behavior. To conduct behavioral experiments with mlSEZt, we first used a split-*Gal4* intersectional approach (*Dionne et al., 2018*) to generate a more specific driver line. We took advantage of a second *Gal4* line that labeled the same bitter-selective mlSEZt neurons as *R29F12-Gal4* (*Figure 7—figure supplement 2A–B*). The split-*Gal4* approach enables us to exclusively label neurons present in both expression patterns, resulting in a highly specific mlSEZt line (*Figure 8A*). We verified that mlSEZt neurons labeled by the split-*Gal4* line respond to bitter (*Figure 7—figure supplement 2C*). This line was used for all behavioral experiments.

We first tested the role of mlSEZt bitter neurons in regulating PER. Activating mlSEZt neurons using Chrimson suppressed PER to sugar (*Figure 8B*), similar to the effect of activating bitter-sensing neurons (*Figure 2A*). Silencing mlSEZt neurons using GTACR reduced the PER suppression elicited by quinine (*Figure 8C*). Thus, mlSEZt neurons partially mediate the aversive effect of bitter taste on proboscis extension. To determine whether this effect translates into a regulation of feeding over longer timescales, we used the optoPAD assay. mlSEZt activation suppressed feeding on sugar (*Figure 8D* and

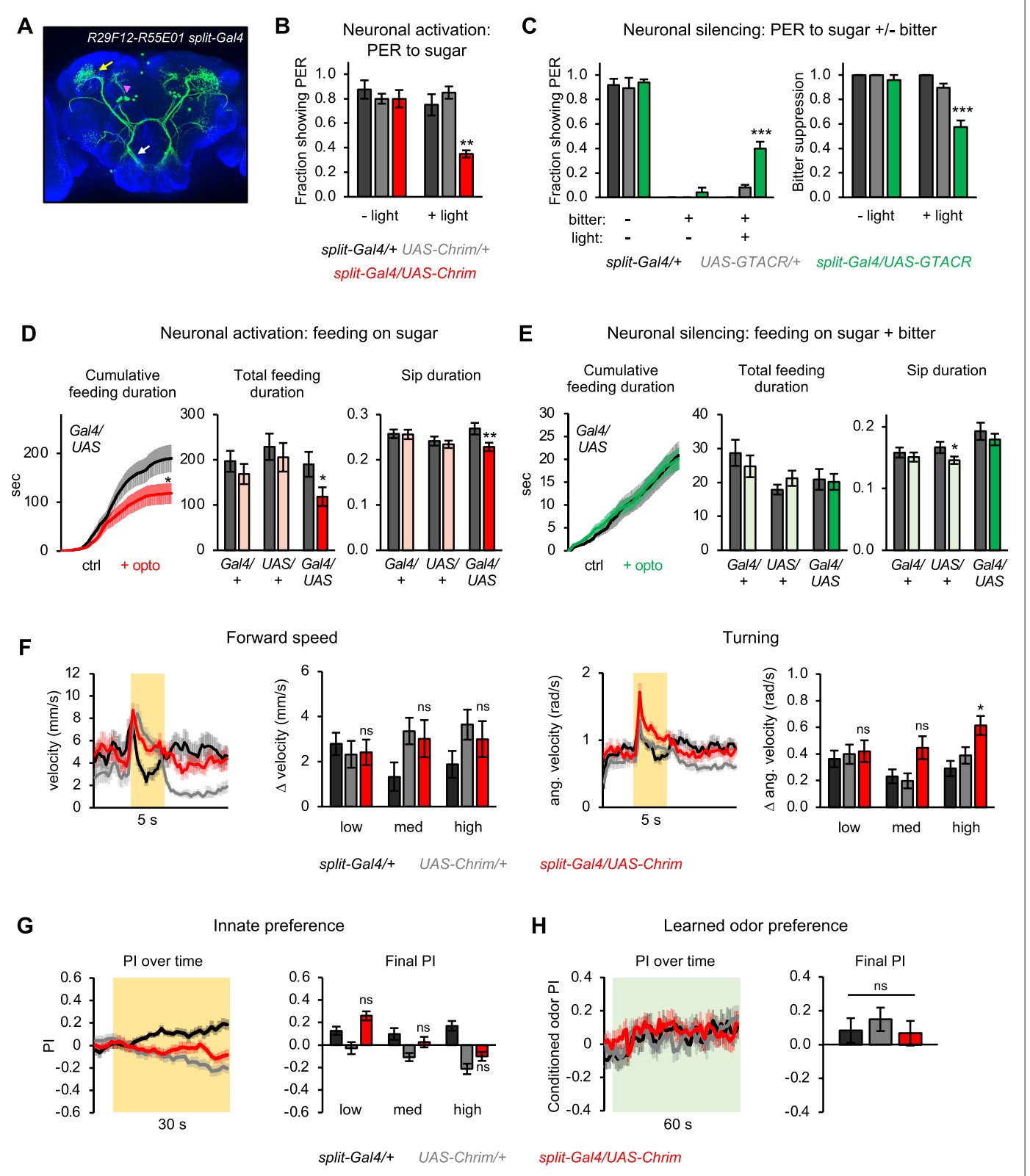

**Figure 8.** Second-order mlSEZt neurons regulate a subset of taste-related behaviors. (**A**) Expression pattern of *split-Gal4* line (*R29F12-AD+R55E01-DBD*) labeling mlSEZt neurons with high specificity. Pink arrowhead shows mlSEZt cell bodies; white and yellow arrows denote their projections in the subesophageal zone (SEZ) and superior lateral protocerebrum (SLP), respectively. (**B–H**) mlSEZt neurons were activated or silenced using *R29F12-R55E01* split-*Gal4* driving *UAS-Chrimson* or *UAS-GTACR*, respectively. (**B**) Effect of activating mlSEZt neurons on proboscis extension response (PER) to 50 mM

*Figure 8 continued on next page*

*Figure 8 continued*

sucrose (n=4 experiments, 40 flies). (**C**) Effect of silencing mlSEZt neurons on bitter suppression of PER, tested by adding 10 mM quinine to 50 mM sucrose (n=5 experiments, 46–50 flies). Left graph shows PER in each condition (3 of the 9 conditions produced zero PER, so bars are at zero with zero error). Right graph shows the degree of bitter suppression based on the left graph, quantified as: 1 – (PER with bitter/PER without bitter). (**D–E**) Effect of activating (D; n=42–47 flies) or silencing (E; n=66–69 flies) mlSEZt neurons on feeding, as measured over 1 hr in the optoPAD (same setup as shown in *Figure 2B and E*, respectively). Food sources contained 100 mM sucrose for activation experiments (**D**) or 50 mM sucrose + 1 mM quinine for silencing experiments (**E**). Left graphs show cumulative feeding duration on the control and opto stim food sources for experimental flies. Values for the last time point (1 hr) were compared using paired t-tests. Cumulative feeding for controls is shown in *Figure 8—figure supplement 1*. Bar graphs on the right represent total feeding duration or sip duration on the control versus opto stim food for each genotype. Paired t-tests were used to compare values for control versus opto stim food. Comparisons not labeled with an asterisk are not significant. (**F–H**) Effects of mlSEZt neuron activation on locomotion (F; n=11–12 trials), innate preference (G; n=22–24 trials, 11–12 sets of flies), and learned odor preference (H; n=14–16 trials, 7–8 sets of flies). Effects of high intensity light stimulation are shown in panels F and G; for other intensities, see *Figure 8—figure supplement 2*.

The online version of this article includes the following figure supplement(s) for figure 8:

**Figure supplement 1.** Additional characterization of mlSEZt effects on feeding behavior.

**Figure supplement 2.** Additional characterization of mlSEZt effects on locomotor and preference behaviors.

**Figure supplement 3.** Effects of silencing mlSEZt neurons on locomotor and preference behaviors elicited by bitter neuron activation.

*Figure 8—figure supplement 1A–B*), consistent with its effect on PER. Certain feeding parameters were suppressed (e.g., total feeding duration, sip duration, number of feeding bouts) while others were not (e.g., number of sips). In contrast to the effect of activation, mlSEZt silencing did not significantly affect feeding on a sugar/bitter mixture (*Figure 8E* and *Figure 8—figure supplement 1C–D*). These results suggest that mlSEZt neurons suppress specific aspects of feeding behavior but act in parallel with other second-order bitter pathways, creating redundancy in the circuit.

In contrast to the effect on proboscis extension and feeding, activating mlSEZt neurons generally did not elicit locomotor changes, innate positional aversion, or learned aversion (*Figure 8F–H* and *Figure 8—figure supplement 2*). The only statistically significant effect in any of these behavioral assays, at any light intensity, was an increase in turning at the onset of high intensity light (*Figure 8F*); medium intensity light also elicited a similar but non-significant effect. Changes in forward velocity at light onset or offset did not differ from controls (*Figure 8F* and *Figure 8—figure supplement 2A–B*). We repeated the locomotor assay at an even higher light intensity (44 µW/mm²) and again observed a significant increase in turning, but not forward velocity (*Figure 8—figure supplement 2D*).

We also tested whether mlSEZt activity is required for bitter-sensing neurons to elicit locomotor responses, innate aversion, or learned aversion. We constitutively silenced mlSEZt neurons by expressing the inwardly rectifying potassium channel Kir2.1 (*Baines et al., 2001*), and we activated bitter neurons across the body by using *Gr66a-lexA* to drive *Aop-Chrimson*. None of the locomotor or preference behaviors elicited by bitter sensory activation were affected by mlSEZt silencing (*Figure 8—figure supplement 3*), indicating that mlSEZt neurons are not required for these behaviors.

Together, these data show that second-order mlSEZt neurons regulate a subset of the aversive behaviors elicited by bitter taste. Behaviors that were not affected by mlSEZt activation may be mediated by other second-order bitter pathways, or they may require the co-activation of multiple second-order pathways.

## Further characterization of mlSEZt cell identity

Having identified a novel type of second-order bitter neuron that regulates aversive behavior, we sought to further characterize its identity and function. The morphology of mlSEZt neurons indicates that they belong to the CREa1 lineage (*Yu et al., 2013*). Neurons within this lineage have sexually dimorphic projections (*Yu et al., 2013*), and mlSEZt neurons share some resemblance with sexually dimorphic *fruitless* (*fru*)-expressing mAL (aDT2) neurons (*Kimura et al., 2005*; *Yu et al., 2010*; *Cachero et al., 2010*), which function in the male courtship circuit (*Clowney et al., 2015*). However, mAL neurons in males do not have a circular projection in the SLP, as mlSEZt cells do. Although our initial characterization of mlSEZt neurons was performed in females (*Figures 7–8*), mlSEZt neurons in males also show a circular SLP projection (*Figure 9—figure supplement 1*), indicating that they are distinct from *fru*+mAL neurons and do not appear to be sexually dimorphic.

We next asked whether mlSEZt cells are excitatory or inhibitory. The CREa1 lineage contains GABAergic neurons (*Ito et al., 2013*), and some neurons resembling mlSEZt in the FlyCircuit database

express *Gad1-Gal4*, a GABAergic marker (e.g., Gad1-F-600213 and Gad1-F-400450) (*Chiang et al., 2011*). To confirm that mlSEZt cells are GABAergic, we colabeled them with the GABAergic marker *VGAT-lexA* and observed co-expression in most mlSEZt cells (*Figure 9A* and *Figure 9—figure supplement 2A*). While some mlSEZt cells did not show clear co-expression, it is possible that expression levels were too low to be detected. To rule out the possibility that some mlSEZt cells are excitatory, we also performed colabeling experiments with the cholinergic marker *ChAT-lexA*, as cholinergic neurons represent the major population of excitatory neurons in the fly brain. We did not observe any mlSEZt neurons that expressed *ChAT-lexA* (*Figure 9A* and *Figure 9—figure supplement 2B*). Thus, mlSEZt bitter neurons are GABAergic cells that convey feedforward inhibition to the SLP.

## Neural circuitry downstream of mlSEZt neurons

Higher-order taste pathways in the superior protocerebrum have not been identified. What downstream circuits do mlSEZt cells connect to? We used *trans*-Tango to label neurons postsynaptic to mlSEZt, which represent third-order bitter neurons (3Ns). Interestingly, mlSEZt neurons themselves were prominently labeled (*Figure 9B*), suggesting that they form connections with each other. We also observed dense staining in the SLP and the neighboring superior intermediate protocerebrum (SIP), with many cell bodies located laterally (*Figure 9B–C*). There were very few projections out of the SLP/SIP, suggesting that 3Ns primarily comprise local neurons. The few projections out of the SLP/SIP mainly arborize in the superior medial protocerebrum (SMP) (*Figure 9C*), the region innervated by second-order bitter neurons comprising the medial tract (*Figure 6*), which suggests potential crosstalk between different second-order pathways. We also observed a medial projection that crossed the midline and innervated the contralateral SLP/SIP (*Figure 9C*). Finally, we observed a single projection that descended vertically from the SMP along the midline, potentially innervating the SEZ, and another projection descending directly from the SLP to the ventral brain (*Figure 9C*).

The dense staining in these *trans*-Tango experiments made it difficult to discern the morphology of single neurons. To identify individual 3Ns downstream of mlSEZt cells, we used the synaptic connectome of the fly brain (the hemibrain connectome; *Scheffer et al., 2020*). Because this connectome does not include the SEZ, we were restricted to analyzing the morphology and connectivity of mlSEZt cells outside the SEZ. We identified 21 neurons whose morphology strongly resembled that of mlSEZt cells (*Figure 9D*). These neurons were classified in the connectome as three different cell types: mAL3A (6 cells), mAL3B (5 cells), and mAL4 (10 cells). All three cell types show a ring-shaped projection in the SLP, with subtle differences in their arborizations (*Figure 9D*). We will refer to these 21 cells as mlSEZt cells, but it is not clear whether bitter-responsive mlSEZt cells include all 21 cells and three cell types, or only a subset. mlSEZt cells show dense SLP arborizations (*Figure 8A*) characteristic of mAL3B or mAL4 cells (*Figure 9D*), but they may also include mAL3A cells with sparser arborizations.

We next used the connectome to identify 3Ns receiving input from these 21 mlSEZt cells. Using a connection threshold of 3 synapses, we identified 201 unique 3Ns. Interestingly, four of the ten 3Ns receiving the strongest mlSEZt input are in fact mlSEZt cells, revealing strong interconnectivity that is consistent with *trans*-Tango labeling (*Figure 9B*). These mlSEZt interconnections show specific connectivity motifs. There are many strong connections from mAL3A to mAL3B, a few connections in the opposite direction (mAL3B to mAL3A), a few connections from mAL3A or mAL3B to mAL4, and almost no connections from mAL4 to the other types (*Figure 9E*). These results suggest a hierarchy where mAL3A strongly influences mAL3B, mAL3A and mAL3B weakly influence mAL4, and mAL4 does not influence the other types. Because these cells are GABAergic, these interactions may mediate inhibitory cross-talk that dampens or gates the responses of other mlSEZt cells.

After excluding interconnections between mlSEZt cells, most of the 3Ns receiving the strongest mlSEZt input are neurons projecting locally within the SLP and SIP (*Figure 9F–H* and *Table 1*), consistent with *trans*-Tango staining (*Figure 9B–C*). Other areas targeted by top 3Ns include nearby regions such as the SMP, lateral horn, superior clamp, and anterior ventrolateral protocerebrum. Very few 3Ns have long-range projections. Different mlSEZt cell types show different connectivity patterns to the top 3Ns (*Figure 9G*). Interestingly, mAL3A cells provide the weakest input to the top 3Ns (*Figure 9G*) despite providing the strongest input to other mlSEZt cells (*Figure 9E*), suggesting a distinction between lateral and feedforward mlSEZt circuitry.

Most of the top 3Ns (*Table 1*) have not been previously studied, but some have been identified in other connectomic studies. Two top 3N types (SLP259 and SLP018) are third-order olfactory

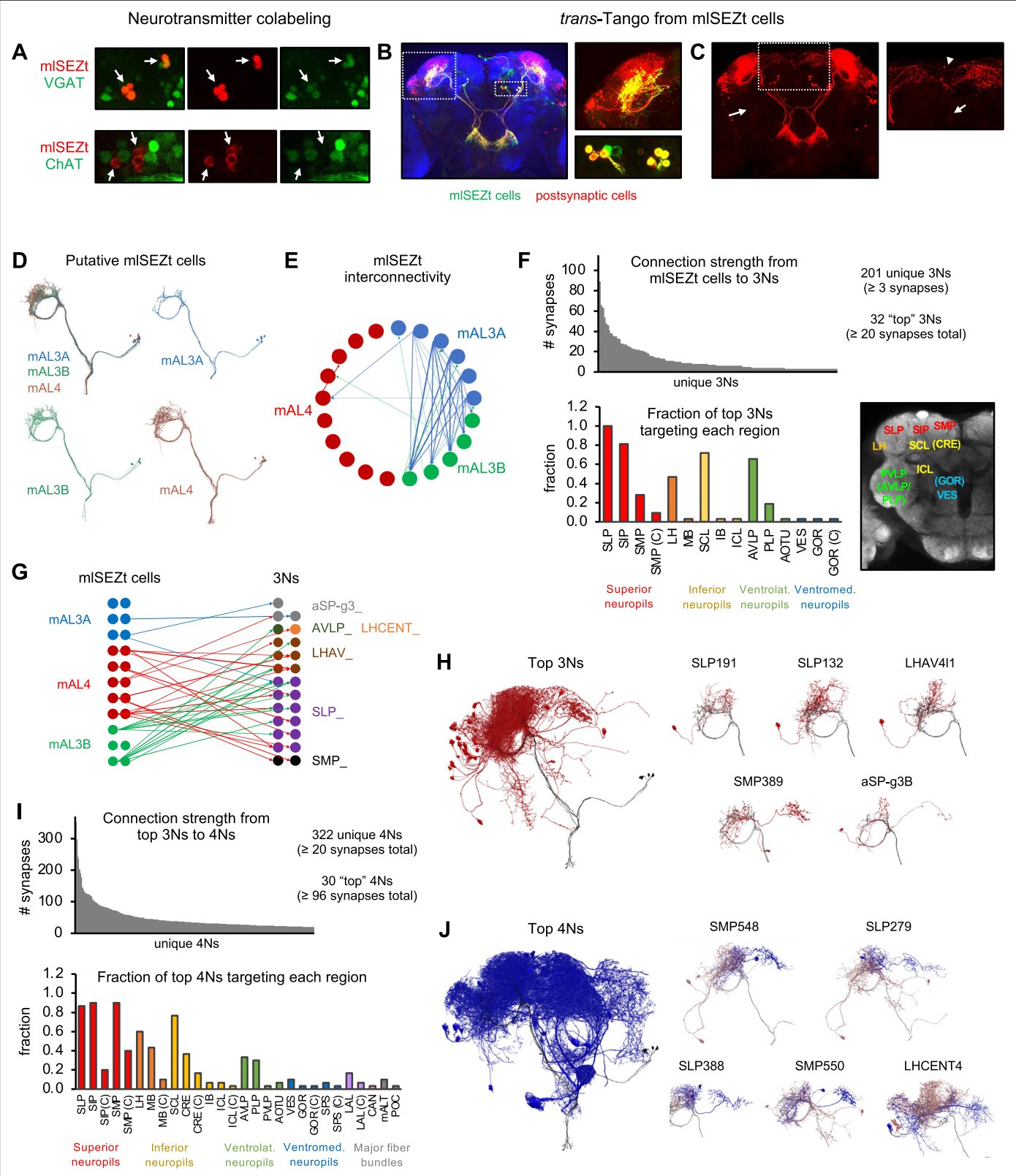

**Figure 9.** Downstream pathways from mlSEZt cells. (**A**) Colabeling of mlSEZt cells with markers for GABAergic (top) or cholinergic (bottom) neurons. Each row shows mlSEZt labeling (red), the neurotransmitter marker (green), and both channels overlaid. Arrows point to the location of mlSEZt cells in each image. Flies contained *R29F12-Gal4* driving *UAS-TdT* (red) and either *VGAT-lexA* (top) or *ChAT-lexA* (bottom) driving *Aop-GCaMP6f* (green). Additional images are shown in *Figure 9—figure supplement 2*. (**B**) Left: *trans*-Tango labeling of cells postsynaptic to mlSEZt neurons (using the

*Figure 9 continued on next page*

*Figure 9 continued*

mlSEZt *split-Gal4* line). Right: Magnified images showing superior lateral protocerebrum (SLP) staining (top) and cell bodies (bottom). Many cell bodies are colabeled in both red and green, representing lateral connections between mlSEZt cells. (**C**) Left: Same image as panel B with only the red channel shown for greater visibility. Arrow shows a ventral projection out of the SLP. Right: Magnified image of the SMP showing a projection to the contralateral hemisphere (arrowhead) and ventral projection toward the SEZ (arrow). For clarity, magnified images in panels B and C represent a subset of the z-slices shown in the whole-brain picture, with brightness and contrast adjusted. (**D**) Putative mlSEZt neurons in the connectome. (**E**) Interconnectivity between mlSEZt neurons. Connections with at least 5 synapses are shown. Line weights represent connection strength (number of synapses; maximum = 30). (**F**) Top: Distribution of connection strength from mlSEZt cells to each 3N. The y-axis represents the total number of synapses each 3N receives from all 21 mlSEZt cells. Bottom left: Fraction of top 3Ns projecting to each brain region. Parenthetical 'C' refers to the specified region in the hemisphere contralateral to the mlSEZt projection. See *Table 1* for abbreviations. Bottom right: Frontal brain slice showing the approximate locations of major regions targeted by 3Ns or 4Ns. Regions in parentheses are not visible in the specific plane shown but are located at that x-y position in an anterior or posterior plane. See *Ito et al., 2014* or NeuPrint Explorer for precise region locations. (**G**) Connectivity between mlSEZt cells and top 3Ns, showing connections with at least 10 synapses (thus not all of the top 3Ns are represented). Line weights represent connection strength (number of synapses; maximum = 18). 3Ns are color-coded based on cell type name prefixes in the connectome, which reflect the cell body location or types defined in previous studies. (**H**) Left: All top 3Ns (red) shown along with four mlSEZt cells (gray). Right: Top five 3N cell types (red) that receive the strongest mlSEZt input, each shown along with three input mlSEZt cells (gray). (**I**) Top: Distribution of connection strength from top 3Ns to each 4N receiving ≥20 synapses. The y-axis represents the total number of synapses from top 3Ns onto each 4N. Bottom: Fraction of the top 30 4Ns projecting to each brain region. (**J**) Left: All top 4Ns (blue) shown along with four mlSEZt cells (gray). Right: Five of the top 4N cell types (blue). Each 4N is shown along with 2–3 input 3Ns (red) and 2–3 mlSEZt inputs to those 3Ns (gray).

The online version of this article includes the following figure supplement(s) for figure 9:

**Figure supplement 1.** Morphology of mlSEZt neurons in male brains.

**Figure supplement 2.** Additional images of mlSEZt colabeling with neurotransmitter markers.

---

neurons in a pheromone-sensing circuit (*Taisz et al., 2022*) and one top 3N, LHCENT1, connects the mushroom body to olfactory circuits in the lateral horn (*Bates et al., 2020*). Several other 3Ns are lateral horn neurons that may participate in olfactory processing (*Schlegel et al., 2021*). Thus, mlSEZt neurons likely provide input to neurons integrating olfactory and gustatory information. The inputs to another top 3N type, aSP-g3, were recently reconstructed using a whole-brain connectome dataset (*Taisz et al., 2022*; *Zheng et al., 2018*). mlSEZt cells (mAL3) as well as several other second-order taste neurons were identified as inputs to aSP-g3, suggesting that some 3Ns integrate input from multiple types of second-order neurons.

Next, we used the connectome to identify postsynaptic partners of 3Ns, which represent fourth-order bitter neurons (4Ns). Starting from the top 3Ns (those receiving ≥20 mlSEZt synapses; 32 cells), we identified over 1700 4Ns total and 322 4Ns that receive at least 20 synapses from the top 3N population (*Figure 9I*). The 4Ns include several mlSEZt cells, revealing feedback connections that likely modulate mlSEZt output. We focused on the 30 4Ns receiving the strongest 3N input, excluding feedback connections to mlSEZt cells (*Table 2*). Most of these top 4Ns send axonal projections to the SLP, SIP, or nearby areas, but collectively they target more diverse brain regions than the 3Ns (*Figure 9I–J*). 4N projections to the SMP are especially prominent, with 90% of top 4Ns sending output to this region. 60% of the top 4Ns project to the lateral horn and 43% project to the mushroom body, regions that mediate innate and learned olfactory processing, respectively. The top 4Ns include neurons that regulate egg-laying (oviIN and oviDNa; two oviDNb cells are also 4Ns but not in the top 30; *Wang et al., 2020*), mushroom body output neurons (*Aso et al., 2014a*), and the PPL201 (PPL2a) dopaminergic neuron (*Li et al., 2020*; *Boto et al., 2019*). The 322 4Ns receiving strong 3N input do not include PPL1 dopaminergic neurons, which convey bitter information to the mushroom body for aversive learning (*Kirkhart and Scott, 2015*), and this may correlate with the inability of mlSEZt cells to elicit learned aversion (*Figure 8H*).

The population of 4Ns also included descending neurons, neurons that project to the ventral nerve cord and regulate body movements such as locomotion. Three of these descending neurons (types oviDNa and oviDNb) drive egg-laying behavior (*Wang et al., 2020*), while a fourth, DNp32, is activated during walking (*Aimon et al., 2022*). Given that mlSEZt activation elicits turning (*Figure 8F*), we searched for downstream connections to DNa02, a descending neuron that controls turning (*Rayshubskiy et al., 2020*). Individual mlSEZt neurons connected to DNa02 via two or three intermediate neurons. We identified many possible connection paths, but a strong and commonly observed path was mlSEZt → SMP389 → AOTU012 → DNa02, with the latter two connections being exceptionally strong (40 and 140 synapses, respectively).

**Table 1.** List of 3Ns receiving at least 20 synapses from mlSEZt cells.

Percent input from mlSEZt refers to the percent of total input synapses the cell receives that come from mlSEZt cells. Target regions for each cell refer to areas where the cell has output synapses. All target regions are ipsilateral to the mlSEZt projections unless specified with '(C)', denoting contralateral innervation. Abbreviations follow NeuPrint conventions: AOTU (anterior optic tubercle), AVLP (anterior ventrolateral protocerebrum), GOR (gorget), IB (inferior bridge), ICL (inferior clamp), LH (lateral horn), MB (mushroom body; includes all lobes and calyxes), PLP (posterior lateral protocerebrum), SCL (superior clamp), SIP (superior intermediate protocerebrum), SLP (superior lateral protocerebrum), SMP (superior medial protocerebrum), VES (vest).

| 3N cell type | Cell ID | # mlSEZt input cells | mlSEZt input types | # mlSEZt synapses | % Input from mlSEZt | Target regions |
|---|---|---|---|---|---|---|
| SLP191 | 420973599 | 7 | mAL3A, mAL3B | 64 | 6.6 | AVLP, SCL, SIP, SLP |
| SLP132 | 359892669 | 6 | mAL3A, mAL4 | 54 | 2.1 | AVLP, LH, SCL, SIP, SLP |
| SMP389 | 575197482 | 8 | mAL3A, mAL3B,mAL4 | 48 | 3.4 | AVLP, PLP, SCL, SIP, SLP, SMP |
| SLP191 | 421313563 | 8 | mAL3A, mAL3B, mAL4 | 47 | 5.0 | AVLP, LH, SCL, SIP, SLP |
| aSP-g3B | 421650982 | 5 | mAL3A | 46 | 12.9 | AVLP, PLP, SCL, SIP, SLP, SMP, SMP(C) |
| LHAV4l1 | 360236724 | 4 | mAL4 | 41 | 3.3 | AVLP, LH, SCL, SIP, SLP |
| SLP179_b | 420623873 | 5 | mAL3B, mAL4 | 38 | 3.7 | SCL, SIP, SLP |
| LHAV2f2_b | 574710121 | 5 | mAL3A, mAL3B, mAL4 | 38 | 7.9 | AVLP, LH, SCL, SLP |
| LHAV2f2_a | 604735525 | 4 | mAL3B, mAL4 | 36 | 9.4 | AVLP, LH, SCL, SLP |
| LHAV2f2_b | 573346324 | 5 | mAL3A, mAL4 | 35 | 5.5 | AVLP, LH, SCL, SLP |
| aSP-g3A | 329919036 | 5 | mAL3A, mAL3B, mAL4 | 35 | 6.3 | SIP, SLP, SMP(C), SMP |
| LHAV1e1 | 390271033 | 4 | mAL3A, mAL3B, mAL4 | 35 | 2.2 | AVLP, LH, SCL, SIP, SLP |
| SLP015_e | 393340402 | 4 | mAL3B, mAL4 | 33 | 6.0 | SIP, SLP |
| SLP011 | 297519736 | 7 | mAL3A, mAL4 | 33 | 2.2 | LH, SCL, SIP, SLP |
| SLP179_b | 391311186 | 5 | mAL3B, mAL4 | 32 | 3.3 | SIP, SLP |
| SLP015_c | 359240144 | 5 | mAL3B, mAL4 | 28 | 3.1 | AVLP, SCL, SIP, SLP |
| aSP-g3B | 485430336 | 5 | mAL3A | 27 | 9.2 | AVLP, SCL, SIP, SLP, SMP |
| SLP187 | 578521941 | 2 | mAL3B | 27 | 11.8 | SIP, SLP |
| SLP187 | 607820937 | 2 | mAL3B | 26 | 7.3 | AVLP, SCL, SLP |
| SLP376 | 298254384 | 4 | mAL3B, mAL4 | 25 | 0.6 | SIP, SLP, SMP |

*Table 1 continued on next page*

*Table 1 continued*

| 3N cell type | Cell ID | # mlSEZt input cells | mlSEZt input types | # mlSEZt synapses | % Input from mlSEZt | Target regions |
|---|---|---|---|---|---|---|
| SLP216 | 5813011119 | 7 | mAL3A, mAL3B, mAL4 | 25 | 1.0 | AOTU, AVLP, GOR, GOR(C), IB, ICL, LH, PLP, SCL, SIP, SLP, SMP, SMP(C) |
| SLP215 | 608534097 | 3 | mAL3B, mAL4 | 24 | 1.1 | AVLP, PLP, SCL, SLP, VES |
| SLP015_c | 359926923 | 4 | mAL3B, mAL4 | 23 | 5.4 | SIP, SLP |
| SLP187 | 483711811 | 3 | mAL3B | 23 | 8.9 | SLP |
| SLP259 | 5813040707 | 4 | mAL3B, mAL4 | 23 | 1.7 | AVLP, SCL, SIP, SLP, SMP |
| AVLP024 | 420965117 | 3 | mAL3B | 22 | 1.2 | AVLP, LH, PVLP, SCL, SIP, SLP |
| LHCENT1 | 328861282 | 4 | mAL4 | 22 | 0.2 | AVLP, LH, MB, SCL, SIP, SLP, SMP |
| LHAV6a10 | 5813047255 | 3 | mAL4 | 22 | 6.5 | LH, SIP, SLP |
| SLP015_e | 329206628 | 2 | mAL4 | 21 | 3.2 | SIP, SLP |
| SMP550 | 452689494 | 3 | mAL3B, mAL4 | 21 | 0.4 | AVLP, LH, PLP, SCL, SIP, SLP, SMP |
| SLP018 | 451663172 | 4 | mAL3A, mAL3B, mAL4 | 20 | 3.1 | AVLP, LH, SCL, SIP, SLP |
| SLP057 | 5813019955 | 4 | mAL3B, mAL4 | 20 | 0.5 | AVLP, LH, PLP, SCL, SIP, SLP |

Together, these analyses suggest that bitter information conveyed by mlSEZt cells is processed in the SLP and SIP before being relayed to other areas, including the SMP, lateral horn, mushroom body, and descending neurons. Strong lateral connections between mlSEZt cells and feedback connections from 3Ns to mlSEZt neurons suggest the presence of extensive local processing. Several 3Ns and 4Ns likely integrate bitter information from mlSEZt with olfactory input, and at least one 3N integrates input from multiple second-order taste neurons. These studies provide the first glimpse into circuits for bitter processing in the higher brain.

## Discussion

In this study, we investigated how different bitter taste inputs are processed in the brain to guide behavior. We took advantage of highly specific genetic tools and a wide range of behavioral assays, representing a systematic and rigorous approach to answering this question. Optogenetic experiments revealed that bitter neurons in different organs elicit broad and highly overlapping effects on behavior, suggesting that these inputs converge onto common downstream pathways. However, we also observed behavioral differences that suggest that these inputs are 'selectively integrated' – downstream pathways may selectively or preferentially receive input from certain bitter neuron subsets. Transsynaptic tracing of second-order pathways supported this interpretation, revealing that bitter neurons in different organs connect to overlapping second-order pathways, but with biased connectivity.

To begin to understand how second-order bitter neurons integrate sensory input and regulate behavior, we characterized a new type of second-order bitter neuron, mlSEZt. We found that mlSEZt neurons receive convergent input from multiple organs and regulate a subset of taste-related

**Table 2.** List of top 30 4Ns.

Top 30 4Ns were selected and sorted based on the total number of synapses from top 3Ns. Percent input from 3Ns refers to the percent of total input synapses the cell receives that come from the top 3Ns. Target regions for each cell refer to areas where the cell has output synapses. All target regions are ipsilateral to the mlSEZt projections unless specified with '(C)', denoting contralateral innervation. Abbreviations not defined in *Table 2*: CAN (cantle), CRE (crepine), LAL (lateral accessory lobe), POC (posterior optic commissure), PVLP (posterior ventrolateral protocerebrum), SPS (superior posterior slope), mALT (medial antennal lobe tract).

| 4N cell type | Cell ID | # 3N input cells | 3N input cell types | # synapses from 3Ns | % input from 3Ns | Target regions |
|---|---|---|---|---|---|---|
| SMP548 | 297580589 | 16 | AVLP024, LHAV1e1, SLP015_c, SLP015_e, SLP057, SLP179_b, SLP191, SLP216, SMP389, SMP550, aSP-g3A, aSP-g3B | 363 | 6.3 | AVLP, LH, PLP, SCL, SIP, SLP, SMP |
| SLP279 | 360591860 | 8 | LHAV1e1, LHCENT1, SLP015_e, SLP018, SLP179_b, SLP216, SMP550, aSP-g3A | 354 | 8.4 | SCL, SIP, SLP, SMP, SMP(C) |
| LHCENT4 | 517506265 | 4 | LHAV4l1, LHCENT1, SLP057, SMP389 | 297 | 4.6 | AVLP, LH,, MB, PLP, SCL, SIP, SLP, SMP, mALT |
| DM1_lPN | 542634818 | 1 | LHCENT1 | 247 | 2.4 | AL, LH, MB, SCL, SLP, mALT |
| LHMB1 | 5813020988 | 2 | LHCENT1, SLP057 | 238 | 3.5 | CRE, LH, MB, SCL, SIP, SLP, SMP |
| SLP388 | 298258611 | 11 | AVLP024, SLP015_e, SLP018, SLP179_b, SLP191, SLP216, SMP389, aSP-g3B | 203 | 3.1 | SCL, SIP, SLP, SMP, SMP(C) |
| SMP550 | 452689494 | 7 | AVLP024, LHAV1e1, SLP018, SLP216, SMP389, aSP-g3B | 201 | 4.2 | AVLP, LH, PLP, SCL, SIP, SLP, SMP |
| oviIN | 423101189 | 2 | SMP389, SMP550 | 190 | 0.8 | CAN, CRE, CRE(C), GOR, IB, LAL, SIP, SIP(C), SMP, SMP(C), SPS, VES |
| LHCENT9 | 330268940 | 10 | AVLP024, LHAV1e1, LHCENT1, SLP132, SLP191, SMP389, aSP-g3A, aSP-g3B | 177 | 1.4 | AOTU, AVLP, LH, MB, SCL, SIP, SLP, SMP |

*Table 2 continued on next page*

*Table 2 continued*

| 4N cell type | Cell ID | # 3N input cells | 3N input cell types | # synapses from 3Ns | % input from 3Ns | Target regions |
|---|---|---|---|---|---|---|
| SLP212 | 5812980529 | 14 | AVLP024, SLP015_e, SLP018, SLP057, SLP132, SLP191, SLP216, SMP389, SMP550, aSP-g3A, aSP-g3B | 145 | 6.7 | SIP, SLP, SMP, SMP(C) |
| SMP108 | 298258513 | 3 | SLP057, SMP389, SMP550 | 145 | 0.6 | CRE, CRE(C), LAL, LH, MB, MB(C), SCL, SIP(C), SIP, SLP, SMP, SMP(C) |
| oviDNa | 550655668 | 2 | SLP216, SMP550 | 137 | 12.3 | CRE, SCL, SIP, SLP, SMP, SMP(C), VES |
| MBON18 | 5813020828 | 1 | LHCENT1 | 132 | 1.0 | LH, MB, SCL, SIP, SLP |
| LHPD4c1 | 421641859 | 9 | LHAV1e1, LHAV4l1, LHCENT1, SLP011, SLP018, SLP057, SLP132, SLP187 | 130 | 2.5 | LH, SCL, SIP, SLP, SMP |
| SLP113 | 390589591 | 6 | AVLP024, LHAV2f2_b, LHAV4l1, LHCENT1, SLP132 | 127 | 10.5 | LH, SIP, SLP, SMP |
| LHPV10b1 | 604709727 | 2 | LHCENT1, SLP057 | 125 | 2.8 | CRE, LH, MB, PLP, PVLP, SCL, SIP, SLP, SMP |
| SMP156 | 673776769 | 3 | SLP216, SMP389, SMP550 | 124 | 2.1 | CRE, GOR(C), IB, ICL(C), ICL, LAL, MB, SMP, SMP(C), SPS, SPS(C) |
| SMP385 | 5813083780 | 2 | SMP389, SMP550 | 124 | 2.0 | CRE, CRE(C), LAL, LAL(C), SCL, SIP, SIP(C), SMP, SMP(C), |
| SLP440 | 328870472 | 13 | LHAV1e1, SLP015_c, SLP015_e, SLP018, SLP057, SLP179_b, SLP191, SLP376, aSP-g3B | 123 | 3.0 | LH, SCL, SIP, SLP, SMP, SMP(C) |
| LHPV5e1 | 328611004 | 5 | LHAV1e1, LHAV4l1, LHCENT1, SLP015_e, SLP132 | 123 | 1.1 | CRE, CRE(C), LH, MB(C), SCL, SIP, SIP(C), SLP, SMP, SMP(C) |
| SMP029 | 604070433 | 3 | SLP216, SMP389, SMP550 | 120 | 6.7 | AVLP, LH, MB, PLP, SCL, SIP, SLP, SMP |
| SMP311 | 5813049378 | 2 | SMP389, SMP550 | 117 | 7.5 | AVLP, ICL, PLP, SCL, SIP, SLP, SMP |

*Table 2 continued*

| 4N cell type | Cell ID | # 3N input cells | 3N input cell types | # synapses from 3Ns | % input from 3Ns | Target regions |
|---|---|---|---|---|---|---|
| PPL201 | 328533761 | 6 | LHAV1e1, LHCENT1, SLP011, SLP015_e, SLP057, SLP179_b | 117 | 1.4 | AVLP, CRE, LH, MB, PLP, POC, SCL, SIP, SLP, SMP, mALT |
| SMP109 | 5813009620 | 2 | SMP389, SMP550 | 107 | 1.7 | AOTU, CRE, CRE(C), LAL, LAL(C), MB, SIP, SMP, VES |
| SMP029 | 541347811 | 4 | SLP215, SLP216, SMP389, SMP550 | 106 | 5.8 | AVLP, MB, PLP, SCL, SIP, SLP, SMP |
| SMP503 | 361312808 | 12 | AVLP024, LHCENT1, SLP011, SLP015_e, SLP018, SLP132, SLP187, SLP215, SMP389, SMP550, aSP-g3B | 102 | 1.5 | AVLP, CRE, LH, MB, PLP, SCL, SIP, SIP(C), SLP, SMP, SMP(C) |
| SLP113 | 421271305 | 8 | LHAV2f2_a, LHAV2f2_b, LHAV4l1, LHCENT1, SLP132, SLP187 | 102 | 10.0 | LH, SIP, SLP, SMP, SMP(C) |
| MBON18 | 457196444 | 1 | LHCENT1 | 97 | 3.6 | LH, MB(C), SCL, SIP, SIP(C), SLP |
| SLP149 | 5813009312 | 9 | LHAV1e1, LHAV4l1, LHCENT1, SLP015_c, SLP018, SLP057, SLP179_b, SLP376 | 97 | 4.3 | SIP, SLP, SMP |
| SLP441 | 5813078542 | 13 | LHAV1e1, LHCENT1, SLP011, SLP015_c, SLP015_e, SLP057, SLP179_b, SLP191, SLP259, SLP376, aSP-g3B | 96 | 3.3 | AVLP, SCL, SIP, SLP, SMP |

behaviors. mlSEZt cells transmit feedforward inhibition to the SLP, and this information is then relayed to several areas including the SMP, lateral horn, and mushroom body. mlSEZt is one of the only types of second-order bitter neurons that has been functionally studied in the adult fly (*Kim et al., 2017*; *Bohra et al., 2018*). Moreover, our studies of downstream circuits from mlSEZt are the first to uncover third- and fourth-order taste pathways in the higher brain.

## Roles of bitter neuron subsets

Conventional models have proposed that taste neurons in different organs have distinct behavioral roles (*Dethier, 1976*; *Scott, 2018*). We were therefore surprised to find that activating bitter neurons within the leg, labellum, or pharynx had broad and largely overlapping effects on behavior. Activating bitter neurons in any of the three organs suppressed proboscis extension and feeding, enhanced locomotor speed and turning, and elicited innate aversion. Neurons in the leg and labellum also elicited learned aversion. These results imply that bitter inputs from different organs converge in the brain to drive a common set of aversive behaviors. However, the effects of different neuronal subsets

were not identical. For example, activating pharyngeal bitter neurons elicited stronger innate aversion than leg or labellar (S-a + I-a) neurons, but the latter two subsets elicited stronger learned aversion than pharyngeal neurons (*Figures 4 and 5*). Differences like these cannot be explained by technical issues such as the number of neurons activated or strength of *Gal4* expression. Together these results suggest a model in which downstream pathways selectively integrate different bitter inputs, enabling specific organs to preferentially regulate certain aspects of behavior (*Figure 5B*).

The strong similarities between the effects of bitter neurons in different organs contrast with studies of sugar taste. Sugar neurons in different organs have different effects on locomotion, feeding responses, and egg-laying (*Thoma et al., 2016*; *Murata et al., 2017*; *Chen et al., 2021*; *Chen et al., 2022*). Moreover, there are specific differences between the roles of sugar and bitter neurons within an organ. For example, pharyngeal bitter neurons strongly regulate proboscis extension (*Figure 2A*), whereas pharyngeal sugar neurons do not (*Chen et al., 2021*). These findings suggest that inputs for appetitive and aversive tastes may be integrated in fundamentally different ways. Perhaps organ-specific roles for appetitive tastes are needed to generate the appropriate sequence of motor actions during feeding, whereas aversive behaviors are more general. It may be adaptive for any organ to be capable of driving nearly any type of aversive response.

We were also curious whether different subsets of bitter neurons within the same organ would show similar or distinct behavioral effects. Due to limitations in driver lines available when we initiated this study, we only compared two such subsets: labellar neurons belonging to the S-a + I-a classes versus the S-b class. S-a + I-a neurons drove aversive effects in all behavioral assays tested, whereas S-b neurons had a more limited effect. However, there are fewer S-b neurons (3) than S-a + I-a neurons (13) and the *Gal4* line used to label S-b neurons seemed to be expressed more weakly, limiting our interpretation of these results. Supporting the notion of class-specific behavioral effects, a previous study showed that two types of pharyngeal bitter neurons regulate different aspects of feeding behavior (*Chen et al., 2019*).

In general, the existence of multiple bitter neuron classes in each organ, with different tuning profiles (*Weiss et al., 2011*), would seem to imply that they have different functional roles. If different classes drive distinct aversive behaviors, this would enable the fly to respond differently depending on which bitter compound is detected. Alternatively, input from different classes may be combinatorially integrated to encode bitter identity. In this case, different compounds could elicit different behavioral responses without each class having a distinct behavioral role.

## Second-order pathways for bitter taste processing

Through transsynaptic tracing, we identified three major second-order pathways that relay bitter input to higher brain areas, which have also been described in other recent studies (*Chen et al., 2019*; *Snell et al., 2022*). Each pathway receives input from multiple organs, although our experiments do not confirm that different inputs converge onto the same individual cells. Labellar bitter neurons connect to all three projection tracts, whereas tarsal bitter neurons primarily connect to the lateral and medio-lateral tracts, and pharyngeal bitter neurons primarily innervate the medial tract. This biased connectivity pattern may provide a neuronal substrate for our behavioral results: different bitter neuron subsets may elicit overlapping but non-identical behavioral effects because they activate overlapping but non-identical second-order taste pathways. For example, bitter neurons in the leg and labellum elicited learning while pharyngeal neurons did not, and this difference may relate to the observation that only leg and labellar neurons connected strongly to the lateral and mediolateral tracts.

Bitter-sensing neurons in different organs show striking differences in their axonal projection patterns (*Wang et al., 2004*; *Kwon et al., 2014*). We were therefore surprised that *trans*-Tango labeling did not reveal any obvious pathways receiving input from just one organ. In the sugar-sensing circuit, several organ-specific pathways have been identified. IN1 interneurons in the SEZ receive sugar input specifically from the pharynx and regulate sugar ingestion (*Yapici et al., 2016*). A set of sugar-responsive serotonergic neurons in the SEZ receives input from the labellum, but not the leg, and regulates sugar intake and insulin release (*Yao and Scott, 2022*). Some tarsal sugar-sensing neurons arborize in the ventral nerve cord (*Kwon et al., 2014*) and activate leg-specific ascending neurons (*Kim et al., 2017*), one of which regulates egg-laying behavior (*Chen et al., 2022*).

The new type of second-order bitter neuron that we characterized, mlSEZt, receives bitter input from both the leg and labellum. This is also the case for the only other second-order bitter projection

neuron that has been previously studied, TPN3 (*Kim et al., 2017*). Bitter inputs from multiple organs are thus integrated at the first synapse, at least within certain second-order pathways. Aside from cross-organ integration, mlSEZt neurons do not show significant transformations of bitter encoding: their taste selectivity, response dynamics, and experience-dependent modulation closely resemble responses in sensory neurons. It will be interesting to compare bitter encoding across different second-order neurons. Different second-order pathways may convey different features of the taste stimulus, perhaps analogous to the parallel 'what' and 'where' pathways in the visual and auditory systems (*Goodale and Milner, 1992*; *Lee, 2013*). For example, different second-order neurons may transmit the bitter ON versus OFF response, thus encoding the start and end of the stimulus. Certain neurons may encode bitter identity while others integrate multiple taste inputs to encode the overall taste valence.

How do individual types of second-order bitter neurons contribute to behavior? Activating mlSEZt neurons revealed that they regulate feeding responses and turning but not locomotor speed, innate preference, or learning. The behaviors not affected by mlSEZt neurons may be regulated by other pathways, and one possibility is that different pathways strictly control different behaviors. This has been suggested in the mammalian taste system, in which a cortico-amygdalar projection regulates licking behavior but is dispensable for reporting the identity of the stimulus in a discrimination task (*Wang et al., 2018*). Similarly, distinct pathways in the olfactory system mediate innate versus learned odor responses (*Bear et al., 2016*; *Amin and Lin, 2019*). Having separate pathways for different behavioral responses allows a circuit to readily modulate individual behaviors depending on context or state.

Alternatively, different second-order pathways may have overlapping behavioral roles. Perhaps different pathways encode different features of the stimulus, as discussed above, and this information is combined downstream in specific ways appropriate for regulating each behavior. In support of this combinatorial model, we found that different bitter-sensing neurons elicited overlapping behavioral effects even when they had largely non-overlapping projection targets. Tarsal bitter neurons mainly project to the lateral and mediolateral second-order pathways, whereas pharyngeal bitter neurons mainly project to the medial pathway, but both subsets strongly affected feeding responses, loco-motor speed, and turning. In addition, activation of either mlSEZt (*Figure 8B*) or TPN3 (*Kim et al., 2017*) suppresses proboscis extension, representing parallel second-order pathways with common behavioral roles. Thus, different second-order pathways may converge downstream to regulate common behaviors. In support of this model, many neurons downstream of mlSEZt (the mediolateral tract) project to the SMP (the target of the medial tract), suggesting potential convergence. More-over, one of the 3Ns downstream of mlSEZt, aSP-g3, receives input from multiple second-order taste neurons (*Taisz et al., 2022*).

## Third- and fourth-order taste pathways

The population of second-order taste neurons has only recently been identified (*Talay et al., 2017*; *Chen et al., 2019*; *Snell et al., 2022*), and almost nothing is known about how taste information is processed further downstream. To gain insight into downstream bitter processing, we traced neural circuits downstream of mlSEZt cells. We first observed that mlSEZt cells connect to each other, and these are mainly unidirectional connections suggestive of hierarchical interactions. Because mlSEZt cells are GABAergic, specific mlSEZt cells likely inhibit other mlSEZt cells and gate their ability to transmit feedforward information. It will be interesting to determine how this motif shapes informa-tion flow, and whether mlSEZt cells representing the sources and targets of these connections have different response properties. Interestingly, the mlSEZt subtype providing the strongest connections to other mlSEZt cells provides the weakest connections to top 3Ns, suggesting a division of labor between lateral and feedforward output from mlSEZt cells.

Different mlSEZt cells also show different connectivity patterns to 3Ns, suggesting that parallel subcircuits may exist downstream of the mlSEZt population. 3Ns represent a broad set of neurons targeting the SLP, SIP, and nearby regions such as the lateral horn. Because mlSEZt neurons are inhibitory, 3Ns should be inhibited by bitter taste and may therefore represent neurons promoting appetitive behaviors. Several 3Ns are likely to process olfactory information, suggesting that they integrate smell and taste inputs. As mentioned above, at least one 3N receives input from multiple second-order taste neurons (*Taisz et al., 2022*). It will be interesting to determine what kinds of

taste input it integrates and whether second-order to 3N convergence is a common feature of the circuit.

While 3Ns mainly target a few brain regions, 4Ns project more broadly. 4Ns include mlSEZt cells, indicating the presence of feedback circuits that may facilitate or dampen mlSEZt transmission. 4Ns include neurons likely to participate in olfactory processing, again suggesting cross-talk between olfactory and taste pathways. Many of the top 4Ns project to the lateral horn and mushroom body, regions that mediate innate and learned olfactory processing, respectively. 90% of top 4Ns project to the SMP, a region that contains neurosecretory cells that regulate feeding (*Nässel and Zandawala, 2020*). 4Ns also include several neurons that regulate egg-laying, suggesting that mlSEZt may convey bitter information to modulate egg-laying decisions. Flies have been shown to display either a preference (*Joseph and Heberlein, 2012*) or aversion (*Dweck et al., 2021*) for laying eggs on bitter substrates, which may depend on the substrate composition or context. The activity of the egg-laying neuron oviDNb, one of three oviDNs in the 4N population, reflects the sugar concentration of the substrate and displays a rise-to-threshold signal that drives egg-laying (*Vijayan et al., 2021*). Thus, oviDNs are poised to integrate multiple types of taste input to control egg-laying behavior.

Aside from oviDNs, the population of 4Ns that we analyzed (those receiving ≥20 synapses from the 3N population; 322 cells) included only one other descending neuron, DNp32. This suggests that additional layers of processing occur before mlSEZt input reaches neurons controlling locomotor behaviors. We identified the turning neuron DNa02 as a fifth-order neuron. While descending neurons control leg and body movements, motor neurons controlling proboscis movements and food consumption are located in the SEZ (*Schwarz et al., 2017*), a region not included in the hemibrain connectome. Future analyses of whole-brain connectome datasets (e.g., *Zheng et al., 2018*) should enable us to identify connections between mlSEZt cells and motor neurons controlling feeding.

## Principles of sensory integration

Within a single taste modality, whether and how different channels of taste input are integrated has been largely unexplored, especially for aversive tastes. Our results show that bitter inputs from different organs are integrated early in the circuit and drive an overlapping, but not identical, set of behaviors. It will be interesting to determine whether similar principles govern the integration of taste input in other taste modalities and other organisms. Humans and other mammals have taste neurons in multiple locations, including three distinct areas of the tongue, the soft palate, pharynx, and internal organs such as the gut (*Liman et al., 2014*; *Depoortere, 2014*; *Travers and Nicklas, 1990*). Moreover, mammalian bitter-sensing cells in the tongue show heterogeneous responses (*Caicedo and Roper, 2001*) and express different combinations of bitter receptors (*Behrens et al., 2007*; *Voigt et al., 2012*), similar to flies, suggesting the presence of different functional classes.

It will also be interesting to compare principles of taste processing and integration with other sensory systems. Like taste, the olfactory system contains well-defined sensory input channels. Because odors activate overlapping ensembles of sensory cells, inputs across the population must be integrated to determine odor identity and behavior, and this begins to occur at the second synapse (*Groschner and Miesenböck, 2019*). However, certain channels of olfactory information may represent 'labeled-line' circuits that are processed separately from other inputs (*Haverkamp et al., 2018*). The mechanosensory system similarly contains well-defined input channels that detect specific stimulus features (e.g., indentation or vibration), suggesting that these channels could potentially mediate distinct aspects of touch perception (*Abraira and Ginty, 2013*). However, recent studies suggest that inputs from different mechanosensory types are integrated, beginning at the first synapse, to create mixed representations (*Emanuel et al., 2021*; *Chirila et al., 2022*). Uncovering when and how sensory circuits integrate different inputs, and how this relates to perception and behavior, lies at the core of understanding how we interpret signals from the world.

# Materials and methods

**Key resources table**

| Reagent type (species) or resource | Designation | Source or reference | Identifiers | Additional information |
|---|---|---|---|---|
| Genetic reagent (*Drosophila melanogaster*) | *Gr33a-Gal4* | *Moon et al., 2009* | BDSC: 31425 | |

*Continued on next page*

*Continued*

| Reagent type (species) or resource | Designation | Source or reference | Identifiers | Additional information |
|---|---|---|---|---|
| Genetic reagent (*Drosophila melanogaster*) | *Gr58c-Gal4* | **Weiss et al., 2011** | BDSC: 57646 | |
| Genetic reagent (*Drosophila melanogaster*) | *Gr59c-Gal4* | **Weiss et al., 2011** | BDSC: 57650 | |
| Genetic reagent (*Drosophila melanogaster*) | *Gr22f-Gal4* | **Weiss et al., 2011** | BDSC: 57610 | |
| Genetic reagent (*Drosophila melanogaster*) | *Gr9a-Gal4* | **Weiss et al., 2011** | BDSC: 57596 | |
| Genetic reagent (*Drosophila melanogaster*) | *R29F12-Gal4* | **Jenett et al., 2012** | BDSC: 49495 | |
| Genetic reagent (*Drosophila melanogaster*) | *R55E01-Gal4* | **Jenett et al., 2012** | BDSC: 39117 | |
| Genetic reagent (*Drosophila melanogaster*) | *R29F12-AD* | **Dionne et al., 2018** | BDSC: 71164 | |
| Genetic reagent (*Drosophila melanogaster*) | *R55E01-DBD* | **Dionne et al., 2018** | BDSC: 69662 | |
| Genetic reagent (*Drosophila melanogaster*) | *Gr66a-lexA* | **Thistle et al., 2012** | BDSC: 93024 | |
| Genetic reagent (*Drosophila melanogaster*) | *VGAT-lexA* | **Deng et al., 2019** | BDSC: 84441 | |
| Genetic reagent (*Drosophila melanogaster*) | *ChAT-lexA* | **Deng et al., 2019** | BDSC: 84379 | |
| Genetic reagent (*Drosophila melanogaster*) | *UAS-Chrimson-TdT* | **Duistermars et al., 2018** | N/A | |
| Genetic reagent (*Drosophila melanogaster*) | *UAS-GTACR1-TdT* | B Noro | N/A | |
| Genetic reagent (*Drosophila melanogaster*) | *UAS-Kir2.1* | **Baines et al., 2001** | BDSC: 6595 | |
| Genetic reagent (*Drosophila melanogaster*) | *trans*-Tango reporter (UAS-Myr-GFP, QUAS-mtdTomato; *trans*-Tango) | **Talay et al., 2017** | BDSC: 77124 | |
| Genetic reagent (*Drosophila melanogaster*) | *UAS-GCaMP6f* | **Chen et al., 2013** | BDSC: 42747 | |
| Genetic reagent (*Drosophila melanogaster*) | *UAS-TdT$^{VK5}$* | D Hattori | N/A | |
| Genetic reagent (*Drosophila melanogaster*) | *UAS-TdT$^{p40}$* | G Rubin and B Pfeiffer | BDSC: 32222 | |
| Genetic reagent (*Drosophila melanogaster*) | *lexAop-GCaMP6f* | D Kim; **Hattori et al., 2017** | BDSC 44277 | |
| Antibody | Anti-GFP (chicken polyclonal) | Aves Labs | Cat# GFP-1020; RRID: AB_10000240 | 1:1000 |
| Antibody | Anti-DsRed (rabbit polyclonal) | Clontech | Cat# 632496; RRID: AB_10013483 | 1:500 |
| Antibody | Anti-bruchpilot (nc82; mouse monoclonal) | Development Studies Hybridoma Bank | Cat# nc82; RRID: AB_2314866 | 1:10 |
| Antibody | Alexa Fluor 488 (goat anti-chicken polyclonal) | Life Technologies | Cat# A11039; RRID: AB_2534096 | 1:500 |
| Antibody | Alexa Fluor 568 (goat anti-rabbit polyclonal) | Life Technologies | Cat# A11036; RRID: AB_10563566 | 1:500 |
| Antibody | Alexa Fluor 633 (goat anti-mouse polyclonal) | Life Technologies | Cat# A21052; RRID: AB_2535719 | 1:500 |
| Software, algorithm | Prism, version 9 | GraphPad | N/A | |
| Software, algorithm | MATLAB | Mathworks | N/A | |
| Software, algorithm | FlyTracker | Caltech; **Eyjolfsdottir et al., 2014** | N/A | http://www.vision.caltech.edu/Tools/FlyTracker/ |

## Fly stocks and maintenance

Flies were reared at 25°C on standard cornmeal or cornmeal-molasses food, with the exception of flies for *trans*-Tango experiments which were reared at 20°. Bitter *Gal4* lines and *UAS-Chrimson* were outcrossed into a *2U* wild-type background for at least five generations. Unless otherwise specified, experiments were performed on mated females, with the exception of *trans*-Tango experiments that used males (the location of transgenes on the X chromosome leads to enhanced expression in males).

Behavioral assays were performed on 3- to 7-day-old flies, calcium imaging was performed on 2- to 3-week-old flies (to ensure strong GCaMP expression), *trans*-Tango staining was performed on ~4-week-old flies (to ensure strong labeling), and other immunostaining experiments were performed on 1- to 2-week-old flies. Flies used for optogenetic experiments were maintained in constant darkness and fed on food containing 1 mM all trans-retinal for 3–5 days prior to testing. PER experiments used 1 day starved flies. optoPAD experiments used flies starved for 1 day (neuronal activation with sweet substrate) or 2 days (neuronal silencing with sweet+bitter substrate). Flies were food-deprived by placing them in an empty vial with a wet piece of Kimwipe containing all 1 mM trans-retinal.

## PER assay

PER was tested using previously described methods (*Devineni et al., 2019*). Briefly, flies were anesthetized on ice, immobilized on their backs with myristic acid, and the two anterior pairs of legs were glued down so that the labellum was accessible for taste stimulation. Flies recovered from gluing for 30–60 min in a humidified chamber. Flies were water-satiated before testing. Tastants were briefly applied to the labellum using a small piece of Kimwipe. Tastant concentrations are specified in the figure legends. For optogenetic stimulation, a red (617 nm) or green (530 nm) LED was manually turned on just before the taste stimulus and remained on for the duration of the stimulus.

Approximately 10 flies were sequentially tested in each experiment, and the percent of flies showing PER to each stimulus was manually recorded. Only full proboscis extensions were counted as PER. At the end of each experiment, flies were tested with a positive control (500 mM sucrose) and were excluded from analysis if they did not respond. For statistical analyses, each set of ~10 flies was considered to be a single data point ('n').

## optoPAD assay

The optoPAD was purchased from Pavel Itskov at Easy Behavior (https://flypad.rocks/), who also provided code to run the assay using Bonsai and to process the data using MATLAB. The design of the optoPAD is described in *Moreira et al., 2019*, and data processing methods are described in *Itskov et al., 2014*. Experiments were performed as described in *Moreira et al., 2019*. Optogenetic activation and silencing were performed using 2.2 or 3.5 V stimulation, respectively. Light onset was triggered immediately upon detection of an interaction with the specified food source, and the light remained on for 1.5 s. Tastants were mixed into 1% agarose.

## Locomotor and preference assays

The behavioral arena for testing locomotor and preference behaviors was built based on designs by the Rubin lab at the Janelia Research Campus (*Aso et al., 2014b*; *Aso and Rubin, 2016*). The arena contains a circular chamber with a glass cover, and flies are filmed from above using a USB camera (Point Grey). The infrared light for illumination and the red (627 nm) LED array for optogenetic stimulation are located just beneath the chamber. Each quadrant of the arena has an inlet for air flow, allowing for odor delivery, and air is removed through a vacuum port at the center of the arena. We did not use air flow for locomotor or innate preference assays. 20–30 flies were tested per trial. Flies were loaded into the chamber using mouth aspiration and were given at least 3 min to habituate before the experiment started. Flies were filmed at 30 frames/s.

For learning experiments using odor, 400 mL/min air was split into two streams that each flowed to one pair of opposing quadrants and could be odorized independently. Each stream was odorized by flowing air through bottles containing odor dilutions. Each stream was then split again to provide input to two opposing quadrants, representing a flow rate of 100 mL/min air into each quadrant. The odors used were 3-octanol (1:1000) and 4-methylcyclohexanol (1:750). Odors were diluted in mineral oil and 2 mL of each odor solution was pipetted onto a piece of thick filter paper (Thermo Fisher Scientific, #88615) placed inside of a glass odor bottle (Cole-Parmer, #EW-99535-16).

The effects of taste neuron activation on locomotion and innate preference were assayed sequentially in the same flies. To quantify locomotor effects, light stimulation was presented for 5 s. Flies were filmed for >1 min before stimulation and 2 min after stimulation, although only the 5 s prior to stimulation was used to quantify baseline behavior. Each of the three light intensities was delivered to the same flies in increasing order of intensity, with several minutes between stimulations to ensure that the behavior had recovered. After an additional rest period, the same flies were then tested for

innate preference by delivering low intensity light stimulation to two opposing quadrants for 30 s. The flies then had a 30 s rest period without light, followed by low intensity light stimulation in the other two quadrants for 30 s. Switching the light quadrants ensured that we could assess light preference independently of any spatial bias. This protocol was then repeated sequentially with medium and high intensity light.

To test learned odor preference, the CS+ odor was presented in all quadrants for 1 min along with light stimulation. Light stimulation started 5 s after odor valve opening and was delivered in 1 Hz pulses, following the protocol used in previous learning studies (*Aso and Rubin, 2016*). After a 1 min rest, the CS- odor was presented alone for 1 min. Following a 1 min rest, the CS+ and CS- odors were simultaneously delivered to different sets of opposing quadrants for 1 min, allowing the flies to choose between the odors. After another 1 min rest, the two odors were presented again for 1 min but the odor quadrants were switched to control for any spatial bias. Which odor was used as the CS+ or CS- was counterbalanced across trials. The light intensity used for learning experiments was 30 µW/mm$^2$.

Fly videos were analyzed using FlyTracker (*Eyjolfsdottir et al., 2014*), which quantified the position, forward velocity, and angular velocity of each fly at each time point. FlyTracker output was further analyzed in MATLAB to quantify preference for light or odor and to generate the traces shown in the figures. Preference index (PI) values were quantified in 1 s bins and calculated as (# flies in light quadrants – # flies in non-light quadrants)/total # flies. PI for the conditioned odor was calculated similarly as (# flies in CS + quadrants – # flies in CS- quadrants)/total # flies. Forward and angular velocities were averaged over 0.33 s bins (10 frames). To quantify locomotor changes at light onset, we averaged forward or angular velocity over the first 1 s of light presentation and compared these values to baseline values averaged over the 4 s preceding light onset. To quantify the change in speed after light stimulation, we averaged forward velocity over a 5 s period following light offset and compared it to the same baseline (pre-light) value described above. For statistical analyses of these data, each trial was considered to be a single data point ('n'). Further analyses are described below in the 'Statistical analyses' section.

## Immunostaining and validation of expression patterns

Immunostaining experiments were performed as previously described (*Devineni et al., 2021*). Briefly, brains were dissected in phosphate buffered saline (PBS), fixed for 15–20 min in 4% paraformaldehyde, washed multiple times with PBS containing 0.3% Triton X-100 (PBST), blocked with 5% normal goat serum for 1 hr, incubated with primary antibodies at 4°C for 2–3 days, washed in PBST, incubated with secondary antibodies at 4°C for 1–2 days, washed in PBST and PBS, and mounted in Vectashield. Primary antibodies used were chicken anti-GFP (1:1000), rabbit anti-DsRed (1:500), and mouse nc82 (1:10). Secondary antibodies used were Alexa Fluor 488 goat anti-chicken (1:500), Alexa Fluor 568 goat anti-rabbit (1:500), and Alexa Fluor 633 goat anti-mouse (1:500). To confirm expression of bitter sensory *Gal4* lines in taste organs, endogenous expression of *UAS-Chrim-TdT* was imaged without immunostaining. Taste organs were removed, immersed in PBS, and imaged within 1–2 hr.

Images for *trans*-Tango staining and characterization of mlSEZt expression (*Figure 6*, *Figure 7A*, *Figure 8A*, *Figure 9B*, *Figure 6—figure supplement 1*, and *Figure 9—figure supplement 1*) were acquired on a Zeiss LSM 710 system. Imaging for validating sensory *Gal4* expression (*Figure 1C*) and neurotransmitter colabeling (*Figure 9A* and *Figure 9—figure supplement 2*) was performed on a Keyence BZ-X810 microscope. All images were processed using Fiji.

## Calcium imaging

Calcium imaging was performed as previously described (*Devineni et al., 2019*; *Devineni et al., 2021*). Flies were taped on their backs to a piece of clear tape in an imaging chamber. For proboscis taste stimulation, fine strands of tape were used to restrain the legs, secure the head, and immobilize the proboscis in an extended position. For leg taste stimulation, the two forelegs were immobilized using tape and parafilm with the distal segment exposed. An imaging window was cut on the anterior surface of the head, the antennae were removed, and the esophagus was cut. The brain was immersed in modified artificial hemolymph in which 15 mM ribose is substituted for sucrose and trehalose (*Marella et al., 2006*; *Wang et al., 2003*).

Images were acquired using a two-photon laser scanning microscope (Ultima, Bruker) equipped with an ultra-fast Ti:S laser (Chameleon Vision, Coherent) that is modulated by pockel cells (Conoptics).

The excitation wavelength was 925 nm. Emitted photons were collected with a GaAsP photodiode detector (Hamamatsu) through a 60X water-immersion objective (Olympus). Images were acquired using the microscope software (PrairieView, Bruker). A single plane was chosen for imaging in each experiment. For most experiments, images were acquired at 256 by 256 pixels with a scanning rate of 3–4 Hz. The resolution was decreased and scanning rate was increased to ~6–7 Hz for experiments using repeated taste stimulation.

Tastants were delivered to the labellum or foreleg as previously described (*Devineni et al., 2019*; *Devineni et al., 2021*). Briefly, we used a custom-built solenoid pinch valve system controlled by MATLAB software via a data acquisition device (Measurement Computing). Solenoid pinch valves (Cole Parmer) were opened briefly (~10 ms) to create a small liquid drop at the end of a 5 µL glass capillary, positioned such that the drop would make contact with the labellum or leg. Tastants were removed by a vacuum line controlled by a solenoid pinch valve. Proper taste delivery was monitored using a side-mounted camera (Veho VMS-004). Five s taste stimulation was used for all experiments. At least three trials of each stimulus were given to each fly. Other than experiments explicitly using repeated taste stimulation, at least 1 min rest was given between trials to avoid habituation. Tastants were used at the following concentrations: 10 mM quinine, 10 mM denatonium, 100 mM caffeine, 25 mM L-canavanine, and 100 mM sucrose.

Calcium imaging data were analyzed using MATLAB code from previous studies (*Devineni et al., 2019*; *Devineni et al., 2021*). Images were registered within and across trials to correct for movement in the x-y plane using a sub-pixel registration algorithm. Regions of interest (ROIs) were drawn manually. Average pixel intensity within the ROI was calculated for each frame. The average signal for 20 frames preceding stimulus delivery was used as the baseline signal, and the $\Delta F/F$ (change in fluorescence divided by baseline) for each frame was then calculated.

## Connectome analysis

mlSEZt neurons in the connectome were identified using NeuPrint Explorer (https://neuprint.janelia.org), an online interface for searching the connectome data. Neuron morphology images in *Figure 9* were generated by NeuPrint Explorer. Connectivity analyses were performed using Python scripts to query the database. 3Ns (201 cells, including mlSEZt cells) were defined as neurons receiving at least three synapses from at least one mlSEZt neuron. Top 3Ns (32 cells) were defined as non-mlSEZt neurons receiving at least 20 total synapses from the mlSEZt population. 4Ns (1743 cells) were identified as neurons receiving at least three synapses from at least one top 3N, but we focused on strong 4Ns that receive at least 20 synapses from the top 3N population (322 cells) and also identified the top 30 4Ns that receive the strongest input (≥96 synapses). Graphs showing the distribution of connection strength (*Figure 9F and I*, top) represent data for all 3Ns (201 cells) or the strong 4Ns (322 cells). Graphs showing the brain regions targeted (*Figure 9F and I*, bottom) represent data for the top 3Ns or 4Ns (32 or 30 cells, respectively). Brain regions can be viewed on NeuPrint Explorer and are based on the nomenclature in *Ito et al., 2014*. The five 3Ns shown in *Figure 9H* represent examples of the top five cell types receiving the strongest total mlSEZt input (based on number of synapses), and they each receive input from at least four mlSEZt cells (see *Table 1*). The five 4Ns shown in *Figure 9J* represent examples of the top cell types that receive the strongest total input from top 3Ns (based on number of synapses) and also receive input from at least three 3Ns (see *Table 2*).

## Statistical analyses

Statistical analyses were performed using GraphPad Prism, Version 9. Statistical tests and results are reported in the figure legends. Two groups were compared using paired or unpaired t-tests. More than two groups were compared using one-way ANOVA followed by Dunnett's test comparing experimental flies to each control. All graphs represent mean ± SEM. Sample sizes are listed in the figure legends. No explicit power analyses were used to determine sample sizes prior to experimentation. Minimum sample sizes were decided prior to experimentation based on previous experience or pilot experiments revealing how many samples are typically sufficient to detect reasonable effect sizes. Additional samples were sometimes added if the initial results were inconclusive or more variable than expected, but never with the intent to make a non-significant p-value significant or vice versa. In general, the experimenter was not explicitly blinded to genotype during experimentation.

To generate the table in *Figure 5A*, the effects of activating bitter neuron subsets were compared to the effect of activating all bitter neurons. For each genotype and behavioral parameter, one-way ANOVAs were used to determine if there was an effect at any light intensity. If so, the strength of the effect was quantified by comparing the maximal effect at any intensity to the maximal effect observed with activating all bitter neurons. To do this, the difference between experimental flies and each control genotype was quantified, and the smaller of the two differences was used as a measure of the experimental effect. A strong or moderate effect was defined as an effect at least 80% or 40%, respectively, of the maximal value. Statistically significant effects that were less than 40% of the maximal value were defined as weak effects.

## Acknowledgements

We thank Richard Axel for his generous support and helpful discussions. We thank Barbara Noro for general advice, sharing unpublished fly lines and code, and providing manuscript feedback. We thank Katie Shakman for the initial setup of the behavioral arena and some analysis code, Tanya Tabachnik and Rick Hormigo for technical assistance with the behavioral arena, Pavel Masek for help with setting up the optoPAD, Jan Hawes for assistance with setting up a new lab, and Deepti Suchindran and Crystal Wang for assisting with pilot experiments in the new lab. We thank Yu-Chieh David Chen and Chris Rodgers for comments on the manuscript. We thank John Carlson, Gilad Barnea, the Janelia Research Center, and the Bloomington Drosophila Stock Center (BDSC) for providing fly strains. This work was supported by internal funding to the Axel lab, startup funding from Emory University (AVD), and a grant from the Whitehall Foundation (AVD).

## Additional information

### Funding

| Funder | Grant reference number | Author |
| --- | --- | --- |
| Whitehall Foundation | 2022-08-017 | Anita V Devineni |

The funders had no role in study design, data collection and interpretation, or the decision to submit the work for publication.

### Author contributions

Julia U Deere, Formal analysis, Investigation, behavioral experiments; brain dissections and immunostaining; Arvin A Sarkissian, Software, Formal analysis, Investigation, Visualization, Writing – original draft, Writing – review and editing, connectome analysis; Meifeng Yang, Formal analysis, Investigation, behavioral experiments; Hannah A Uttley, Formal analysis, Investigation, behavioral experiments; brain dissections and immunostaining; Nicole Martinez Santana, Investigation, behavioral experiments; Lam Nguyen, Formal analysis, Investigation, behavioral experiments; Kaushiki Ravi, Formal analysis, Methodology, behavioral experiments; Anita V Devineni, Conceptualization, Data curation, Software, Formal analysis, Supervision, Funding acquisition, Validation, Investigation, Visualization, Methodology, Writing – original draft, Writing – review and editing, brain dissection and immunostaining; calcium imaging

### Author ORCIDs

Arvin A Sarkissian http://orcid.org/0000-0002-6663-2989
Hannah A Uttley http://orcid.org/0000-0003-0573-6546
Anita V Devineni http://orcid.org/0000-0001-9540-8655

### Decision letter and Author response

Decision letter https://doi.org/10.7554/eLife.84856.sa1
Author response https://doi.org/10.7554/eLife.84856.sa2

## Additional files

### Supplementary files
• MDAR checklist

### Data availability
Source data files for experiments and code for connectome analyses are publicly available at https://github.com/avdevineni/taste-activation, (copy archived at swh:1:rev:920e0cd3d5d03b02e30faa3129e0d2d6b7dad8d6).

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
