## [Editor Report]

This important manuscript addresses the complexity of processing and representation within bitter taste perception using the *Drosophila* model. The authors provide convincing experimental support for distinct anatomical pathways that process bitter tastes and converge on joint downstream neurons to elicit avoidance responses. The combination of behavioral assays, in vivo physiology, optogenetic manipulation, and connectomics leads the authors to a compelling model of bitter taste processing.

---

## [Decision Letter]

**Decision letter after peer review:**

[Editors’ note: the authors submitted for reconsideration following the decision after peer review. What follows is the decision letter after the first round of review.]

Thank you for submitting the paper "Selective integration of diverse taste inputs within a single taste modality" for consideration by *eLife*. Your article has been reviewed by 3 peer reviewers, one of whom is a member of our Board of Reviewing Editors, and the evaluation has been overseen by a Senior Editor. The following individual involved in the review of your submission has agreed to reveal their identity: Alex C Keene (Reviewer #2).

Comments to the Authors:

We are sorry to say that, after consultation with the reviewers, we have decided that the work in its current form will not be considered further for publication by *eLife*.

While the reviewers agree that the topic of the study is of general interest, they also felt that the work does not reach the level of significance and novelty expected for an *eLife* article. Specifically, the reviewers find that the study is divided into two parts. The first part is overall convincing, but does not offer sufficient advances beyond the current understanding and literature in the field. The second part has potential and contains some very interesting data. Unfortunately, the reviewers felt that the data of this part are too preliminary at this point. The reviewers discussed several possibilities to help you improve your study given its potential. First, they recommend shortening and re-organizing the manuscript and center it on part two of the study. Second, they recommend strengthening the second part with additional experiments and controls. We hope that the detailed reviews below will be helpful.

*Reviewer #1:*

Deere et al. combine optogenetics, behavioral analysis and calcium imaging methods to carefully study the effects of bitter neuron activation on different body parts on behavior. While activation of bitter neurons in different organs drives mostly the same behavior, they find a few subtle differences in behavioral output upon neuronal activation between pharynx and labellum, for instance. Using trans-tango labeling and a split-Gal4 line the authors aim to explain their findings. The data are of high quality and contain interesting information for people in the field.

Line 155: The authors state that sugar neuron activation drives a stronger behavioral effect in starved as compared to fed flies. Given that the authors are comparing the two systems, it would be nice to add a possible explanation. For instance, this would suggest that hunger state modulates behavioral responses to sugar later in processing, but this is not/less the case for bitter.

Line 196: it would be interesting to hear and include the authors' interpretation of this result and what it says about the underpinning circuits.

Line 217ff: these are nice results and confirm previous observations, but it is unclear how far they support or fit into the narrative. I encourage the authors' to consider removing this part to make the paper a bit leaner and straight forward, and to present it only in the context of the next paragraph.

Line 241ff: the data indicates that the subset that diverges most is the labellar subset S-b. The authors talk about differences in expression levels and are careful to not compare the subsets directly. On the other hand, they plot the data in the same graphs so that one automatically compares them, but ok. I am a little worried about cell number. You are activating 13/19 vs. 3/19. As you state in the discussion, these number differences could explain the behavioral effects rather than being of different subtypes. To make it easier for the reader, it might be better to move the S-b data into the supplements and remove them from the comparison in the main figure to essentially compare legs vs. labellum vs. pharynx. If you carry out this comparison, then it's evident that the only difference is seen for learned odor preference (pharynx activation does not induce it). This makes perfect sense looking at the trans-tango data (no connections to higher brain centers, which I find very interesting).

Line 299ff: the trans-tango data are very nice, but have the authors considered integrating available EM data, e.g. from Neuprint, into their study? Such data could be used to trace the circuits further downstream. It would also help to test their main conclusion with other means, i.e. projection neurons integrate bitter taste from multiple organs. Is this true for other bitter/taste projection neurons?

Discussion:

It is surprising that the activation of the bitter projection neuron does not induce learning. What are the connections of these neurons in higher brain regions?

Your data are of very high quality and very convincing. I am, however, not quite convinced by the interpretation.

Looking at the data, I see that pharynx induces different behaviors (in particular learning) than the other subsets. Your trans-tango data provides a possible explanation. Your other data suggest that, indeed, leg and labellum drive very similar behavior (with perhaps the exception of 'post-light locomotion). This also fits to the trans-tango data overall.

Therefore, it is difficult for me to follow your conclusion, e.g. abstract: 'These results suggest that different bitter inputs are selectively integrated early in the circuit, enabling the pooling of sensory information, and the circuit then diverges into multiple pathways that may have different behavioral roles.'

Given the high quality of your data, I suggest condensing the manuscript and perhaps focussing on the differences between pharynx and the other neurons a little more. Moreover, using more inactivation experiments could be helpful, too, to distinguish the differences between these subsets in the different behaviors you study. Finally, the paper is divided into two parts, and I am not convinced that the sugar neuron data is needed for the second part and vice versa. It might make sense to split the manuscript into two.

*Reviewer #2:*

The authors tackle a particularly interesting question of how multiple taste cues from the same modality are integrated to form a behavioral response. The manuscript is scientifically sound, with well controlled experiments. The authors rely on optogenetic manipulations to define behavioral responses to different subsets of neurons. The field has tended to rely on relatively simple behavioral responses, such as PER, and a strength of this manuscript is the detailed characterization of locomotor behavior, learning, and reflexive feeding. This systematic characterization of bitter responsive neurons will be broadly useful to the field. In addition, the identification of second order neurons that selectively regulate PER provides a platform for studying integration. A number of concerns are noted that are readily addressable. First, there is a reliance on optogenetics that may not reflect the temporal or spatial patterns of natural compounds. The impact of the paper may be greater if these experiments were complemented with naturally occurring compounds. Second, as written much of the manuscript is descriptive, focusing on aspects that have previously been explored (such as the opposing valances of bitter and sugar neurons). Streamlining this part of the manuscript will make the significance more accessible. Overall, this manuscript provides a valuable resource for understanding how bitter tastants are coded, and will be of broad interest to groups researching sensory processing and behavioral regulation.

1. The paper focuses on the complexity of taste coding, however, it does not firmly establish why the optogenetic approach is needed. There are many limitations to this approach, including the fact that the subsets of neurons, and their temporal activation may not reflect those induced by food. Additional justification of the use of this system is important.

2. The conclusions of Figure 2 are unclear. One possibility is that the activation of bitter neurons induces locomotion, which results in less time on those quadrants. The other is that they are actively avoiding the quadrant. Without addressing this, it is not clear what this adds to the conclusions beyond the data presented in Figure 1.

3. The conclusion of the first four figures are factors that sweet and bitter neurons have oppositional valance, and that sugar is more potentially modulated by a huger state than bitter taste. These have been well described throughout the literature in multiple systems. At the authors' discretion, I suggest streamline these as it will make the novel aspects of the manuscript more accessible.

4. Are the GAL4 lines used to characterize distinct subsets of neurons exclusively expressed in these body regions or preferentially expressed in them? I presume the former, but it would be valuable to reference this and/or show expression if it has not been thoroughly described across all regions.

5. It would be helpful to directly compare subsets of sweet and bitter second order neurons in figure 7 to gain insight into whether the circuit design is shared between the tastants.

Recommendations for the authors:

1. In numerous cases the justification for the experimental approach is unclear. For example line 246 states 'We focused on bitter taste for these studies.' It would be helpful to know why the bitter taste was chosen over sweet.

2. The introduction is very well written and frames the question of how tastants are detected. However, the focus primarily on the sweet and bitter modalities, and discusses how these are sensed by Grs. The complexity described includes the more recent descriptions of other modalities and new classes of gustatory receptors. It seems important to describe these as well in order to convey how the complexity of the system that is being dissected.

3. Line 43 says 'like other sensory systems…' but the focus of the introduction is almost exclusively on taste. Much of this can be streamlined, and it would be helpful to add comparisons for context. For example, questions of sensory integration and combinatorial receptor activation have been addressed in much greater detail for olfaction and this may be worth mentioning.

4. Line 109: It's not clear what the first sentence means. Most tastes are likely to be feeding related. Is the point that most studies have used taste response to food components?

*Reviewer #3:*

This manuscript by Deere et al. characterizes the behavior effects of optogenetic activation of sugar and bitter sensing GRNs. While activation of sugar-sensitive GRNs causes a brief decrease in locomotion, activation of bitter-sensitive GRNs increases locomotor activity in flies. Furthermore, the impact of sugar-GRN activation, but not bitter-GRN activation, is enhanced by food deprivation. The authors, also showed, pairing neutral odors with optogenetic activation of sugar-GRNs or bitter-GRNs can induce positive or negative valence to neutral odors in preference experiments in subsequent trials. After characterizing behavioral effects of optogenetic manipulation of sugar and bitter GRNs, the authors next focus on activation of bitter-sensing GRNs in different organs and discuss the differences and similarities between their behavioral effects. The authors observe some overlap between the behavior effects elicited by bitter-sensing GRN activation in different organs, legs, pharynx, labellum, etc. therefore they hypothesized bitter GRNs might share the same downstream neural circuits. Using transsynaptic labeling, they identified a group of neurons that receive input from different bitter sensing neurons, the mlSEZt neurons, and they showed that mlSEZt neurons showed ON and OFF responses to bitter tastant s but not sugar using calcium imaging. Interestingly, the response dynamics of mlSEZt neurons show different responses to bitter compounds, and the ON response of mlSEZt neurons habituated faster than the OFF response in response to repeated stimulation.

Overall, this is a promising study, however, I have some major technical concerns about the way experiments have been conducted. Without having some clarification of GRN activity during optogenetic activation and anatomical characterization of the GRN drivers used in the study, it is hard to interpret what the results mean.

1. In all of the optogenetic activation experiments, the authors are using constant light stimulation. Previous papers (Inagaki 2014, Nature Methods) have shown that constant light stimulation causes suppression of action potentials in GRNs. Hence, it is not clear how the stimulus protocols used in the manuscript are impacting GRN activity. Authors need to show that either GRNs are actually activated and not shut down by the prolonged optogenetic stimulation protocols they used, via electrophysiology or calcium imaging. Alternatively, they can change their activation protocols by shortening the stimulus period or using pulses of light instead of constant light activation. Without having some evidence, the GRNs are actually active during the periods of optogenetic activation, it is hard to interpret the results of the experiments in Figure 1-5.

2. For sugar-GRN activations, it would be good to use another sugar-GRN, such as Gr5a to confirm Gr64f results.

3. In figure 5, the authors are activating subsets of bitter-GRNs and measuring the effects of optogenetic stimulations in various behaviors. I would like to see the expression patterns of GRN drivers used in these experiments in flies expressing Chrimson. Chrimson transgene used in these experiments has a Td-tomato tag, so authors can stain for this transgene and show us exactly which neurons they are stimulating. Without detailed anatomical characterization of the driver lines used in the experiments, it is again hard to interpret the results.

4. See comment #2. The same concern here as Figure 5. Please show the expression pattern of the source GAL4s that are used for transsynaptic labeling. Without knowing the GRN expression, we can't interpret the results of downstream neurons.

5. In figure 8, are the ON and OFF responses of mlSEZt neurons displayed by arbors of the same neurons, or different neurons? I.e., is there any internal difference between different mlSEZt neurons, in terms of their response dynamics to bitter compounds?

Recommendations for the authors:

1. In Figure 8 – —figure supplement 1C and D, it is better to include a diagram encircling the region of ROI in the brain. The arrows in Figure 8D are unclear.

2. For the experiments that involved optogenetic activation of both sugar sensing and bitter sensing neurons, how could you know that the behavior effect is not influenced by the possible different GAL4 expression levels of sugar sensing and bitter sensing GAL4 lines?

3. In Figure 1 – —figure supplement 1, there should be '(D)' in the last figure.

4. In Figure 8, adding experiments with Gr66a-lexA > CsChrimson, mlSEZt-Split (the cleanest line) > TNT might be helpful in figuring out whether mlSEZt neurons are necessary for the locomotor responses, innate and learned preference. Also, as a negative control, Gr5a-lexA > CsChrimson, mlSEZt-Split > TNT can be tested if reagents are available.

5. In Figure 8G, angular velocity significantly increases with high led intensity. Would it be possible to use higher led intensity to make sure that all mlSEZt populations are activated?

[Editors’ note: further revisions were suggested prior to acceptance, as described below.]

Thank you for resubmitting your work entitled "Selective integration of diverse taste inputs within a single taste modality" for further consideration by *eLife*. Your revised article has been evaluated by Claude Desplan (Senior Editor) and a Reviewing Editor.

The manuscript has been improved but there are some remaining issues that need to be addressed, as outlined below:

Please address all reviewer comments with a detailed response and/or changes to the manuscript text and figures. If you have the data available for points 6 and 7 of Reviewer 2, please include them. Otherwise, please change the text to address the fact that this driver is not optimal.

*Reviewer #1 (Recommendations for the authors):*

The authors have done a commendable job in revising and thereby strongly improving the manuscript. As previously stated, the data is of high quality, and the results are convincing. The new results including the connectomics analysis are very interesting and integrate the findings in early sensory processing with higher-order processing. Moreover, the new optoPAD data including neuronal inhibition is helpful to make the point of functional redundancy and, in combination with the later results, highlights a difference in redundancy at different levels of behavioral performance and learning.

The authors have addressed my previous concerns related to the first submission of this work in full. I am satisfied with their responses. In particular, I believe the paper in its current, more streamlined form is easier and more interesting to read after the removal of some of the data including the data on sugar sensing.

*Reviewer #2 (Recommendations for the authors):*

My previous comments have been addressed. Here are some issues that still remain.

Issues to be addressed in the revised version.

1. The images are too small in Figure 1C. I suggest making three columns: One for the labellum only, one for the legs only, and one for the pharynx only. For the labellum column, authors can zoom in to the labellum, and for the pharynx, they can only zoom in to the pharynx. It will make the results in 1D clearer. As it is, neurons are not clearly visible.

2. Figure 2D, I don't see the point of plotting opto-ctrl and getting negative values; I think it is easier to plot ctrl-opto and plot positive values rather than negative values. Nothing will change, except the plots will look better, in my opinion. Right now, it is confusing to see negative values. Same argument for Figure 2G. Also, some of the negative controls seem to have a high variance. Did the authors test for ANOVA and compare all controls and test animals with each other statistically? If yes, please indicate statistical significance in the plots.

3. I think Figure 6 is missing the VNC expression. Many leg neurons project to the VNC, showing only the brain Trans-tango signal is misleading.

4. I don't think there is any reason for 9 figures. Figures 5-6 can be combined or shown as supplementary figures.

5. I think it is too strong to say mlSEZt neurons are second-order bitter neurons. There is no experiment showing direct synaptic connectivity to bitter neurons. They can say bitter downstream neurons or show synaptic connectivity via EM or GRASP.

6. Authors use a 'dirty' GAL4 for functional imaging experiments rather than the 'clean' split line. I think it might be better to use the cleaner split-GAL4 for functional imaging since many neurons are labeled in SEZ and hard to know which neurons' projections are recorded during imaging.

7. Effects on behavior upon silencing or activating the clean split are much weaker. I think functional imaging is essential to demonstrate these are the real downstream bitter neurons.

8. Figure 9 is interesting, but it is not super relevant without really demonstrating whether mlSEZt neurons respond to bitter compounds. Also hard to demonstrate, the putative mLSEZt neurons they found in EM data are really mLSEZt neurons.

*Reviewer #3 (Recommendations for the authors):*

This revised manuscript has been significantly reorganized to focus on bitter taste. The streamlining of results and inclusion of clarifying data significantly strengthen this manuscript, and further our understanding of bitter taste processing. With these changes, the authors have addressed all of my previous concerns.

---

## [Author Response]

[Editors’ note: the authors resubmitted a revised version of the paper for consideration. What follows is the authors’ response to the first round of review.]

While the reviewers agree that the topic of the study is of general interest, they also felt that the work does not reach the level of significance and novelty expected for an eLife article. Specifically, the reviewers find that the study is divided into two parts. The first part is overall convincing, but does not offer sufficient advances beyond the current understanding and literature in the field. The second part has potential and contains some very interesting data. Unfortunately, the reviewers felt that the data of this part are too preliminary at this point. The reviewers discussed several possibilities to help you improve your study given its potential. First, they recommend shortening and re-organizing the manuscript and center it on part two of the study. Second, they recommend strengthening the second part with additional experiments and controls. We hope that the detailed reviews below will be helpful.

We understand the reviewers’ perspective, and we have completely reworked the manuscript to address their comments and build a stronger story. This includes removing many experiments that the reviewers did not think were sufficiently novel or relevant, and adding many new results to strengthen and expand the story. We have made the following major revisions:

1) The paper now focuses entirely on the bitter pathway; all experiments with sugar neurons have been removed. Moreover, the paper focuses from the very beginning on the core question of how different bitter inputs are processed. Figures 1-4 of the original paper no longer exist in their previous form.

2) We have conducted new experiments using a high-resolution feeding assay (the optoPAD). Previously we only used proboscis extension as a simple readout of feeding initiation. The new experiments are important for testing more naturalistic feeding behavior in freely moving flies, as well as analyzing detailed parameters of feeding.

a) We activated bitter neuron subsets, and the results were consistent with our previous data using proboscis extension: all subsets strongly suppressed feeding, with certain feeding parameters affected more than others (Figure 2B-D).

b) We silenced bitter neuron subsets in the presence of natural bitter taste, which some reviewers asked for. In contrast to the activation experiments, silencing generally did not produce statistically significant effects compared to genetic controls (Figure 2E-G). Together with the activation data, these results support a model in which different bitter neuron subsets act in a parallel and largely redundant manner to regulate aversive behaviors.

c) We activated or silenced second-order mlSEZt neurons (Figure 8D-E). mlSEZt activation suppressed feeding, and we observed effects on certain feeding parameters (e.g. sip duration) but not others (e.g. number of sips). mlSEZt silencing did not have a significant effect. These findings suggest that mlSEZt cells suppress specific aspects of feeding behavior but act in parallel with other second-order bitter pathways, creating redundancy in the circuit.

3) We have presented control experiments to address concerns about the Gal4 lines and neuronal activation protocols.

a) Reviewers 2 and 3 asked about the specificity of Gal4 lines targeting bitter neuron subsets. We now emphasize that all of these Gal4 lines have been extensively characterized in previous studies, and we show our own images confirming their organ-specific expression (Figure 1C).

b) We have added new experiments to address concerns regarding the stimulation protocol, described further in our responses to specific reviewer comments (see below).

4) Given the reviewers’ interest in the second part of the paper, which focuses on downstream bitter pathways, we have expanded this part of the study. We reorganized the figures to bring some of the supplemental imaging experiments into the main figures, as they provide insight into how mlSEZt encodes taste, and we have added many new results:

a) We performed additional behavioral experiments with mlSEZt suggested by Reviewer 3. First, we tested the effect of higher intensity activation on locomotion, which revealed similar effects as the lower intensities. Second, we silenced mlSEZt while activating bitter-sensing neurons. We found that mlSEZt is not required for bitter neurons to affect locomotion, preference, or learning, as expected from the inability of mlSEZt activation to elicit these behaviors.

b) We used co-labeling experiments to show that mlSEZt cells are GABAergic, revealing that they transmit feedforward inhibition to the SLP. This has implications for the circuit analysis below.

c) We traced the neural circuits downstream from mlSEZt neurons using two methods, *trans*Tango and connectome analysis. These studies revealed that bitter information from mlSEZt cells is first processed in the SLP and SIP before being relayed to other areas, including the SMP, lateral horn, mushroom body, and descending neurons. There are strong interconnections between mlSEZt cells and feedback connections from third-order neurons to mlSEZt cells, suggesting extensive local processing. Several downstream neurons are poised to integrate bitter information with olfactory input, and at least one neuron integrates input from different second-order taste neurons. There are also connections to oviDN neurons that use sweet taste input to regulate egg-laying decisions, suggesting that they integrate sugar and bitter inputs. These studies are the first to uncover third- and fourth-order taste pathways in the higher brain.

Overall, we believe that these revisions have resulted in a much stronger manuscript that should reach the level of significance expected for *eLife*. It is striking how little is known about central taste processing in flies, and our study provides several new insights that will have broad interest to the field. This includes: (1) discovering that bitter neurons in different organs drive largely overlapping behaviors, which challenges the conventional model, (2) revealing principles of taste circuit connectivity at the first synapse, (3) characterizing the response properties and behavioral role of a new type of second-order bitter neuron, which remains one of the only second-order taste neurons to be studied, and (4) revealing circuits for taste processing in the higher brain.

Reviewer #1:Deere et al. combine optogenetics, behavioral analysis and calcium imaging methods to carefully study the effects of bitter neuron activation on different body parts on behavior. While activation of bitter neurons in different organs drives mostly the same behavior, they find a few subtle differences in behavioral output upon neuronal activation between pharynx and labellum, for instance. Using trans-tango labeling and a split-Gal4 line the authors aim to explain their findings. The data are of high quality and contain interesting information for people in the field.Line 155: The authors state that sugar neuron activation drives a stronger behavioral effect in starved as compared to fed flies. Given that the authors are comparing the two systems, it would be nice to add a possible explanation. For instance, this would suggest that hunger state modulates behavioral responses to sugar later in processing, but this is not/less the case for bitter.

We have removed all results related to sugar neuron activation and fed vs. starved comparisons, so this comment no longer applies.

Line 196: it would be interesting to hear and include the authors' interpretation of this result and what it says about the underpinning circuits.

This comment refers to results with sugar neuron activation, which have been removed from the paper.

Line 217ff: these are nice results and confirm previous observations, but it is unclear how far they support or fit into the narrative. I encourage the authors' to consider removing this part to make the paper a bit leaner and straight forward, and to present it only in the context of the next paragraph.

We have removed these results from the paper.

Line 241ff: the data indicates that the subset that diverges most is the labellar subset S-b. The authors talk about differences in expression levels and are careful to not compare the subsets directly. On the other hand, they plot the data in the same graphs so that one automatically compares them, but ok. I am a little worried about cell number. You are activating 13/19 vs. 3/19. As you state in the discussion, these number differences could explain the behavioral effects rather than being of different subtypes. To make it easier for the reader, it might be better to move the S-b data into the supplements and remove them from the comparison in the main figure to essentially compare legs vs. labellum vs. pharynx. If you carry out this comparison, then it's evident that the only difference is seen for learned odor preference (pharynx activation does not induce it). This makes perfect sense looking at the trans-tango data (no connections to higher brain centers, which I find very interesting).

We agree that the data would be cleaner if we ignore S-b activation, but we think it would be misleading to relegate that data to the supplemental figures without a strong rationale. We do not have strong evidence that the weak S-b effects are solely due to the low cell number; for example, pharyngeal activation using *Gr9a-Gal4* only labels one cell and yet had a very strong effect on many behaviors.

To draw conclusions about the functions of different organs (e.g. in the text and Figure 5B model), we are considering the S-a + I-a labellar data rather than the S-b data, just as the reviewer suggests. We think this is justified because positive results are more informative than negative results.

We agree that the difference in behavior and connectivity for the pharynx versus the other organs is interesting. We have added a sentence to make the point that the reviewer mentions (lines 636-638): “For example, bitter neurons in the leg and labellum elicited learning while pharyngeal neurons did not, and this difference may relate to the observation that only leg and labellar neurons connected strongly to the lateral and mediolateral tracts.”

Line 299ff: the trans-tango data are very nice, but have the authors considered integrating available EM data, e.g. from Neuprint, into their study? Such data could be used to trace the circuits further downstream. It would also help to test their main conclusion with other means, i.e. projection neurons integrate bitter taste from multiple organs. Is this true for other bitter/taste projection neurons?

The NeuPrint (hemibrain) connectome, which is the only publicly available annotated connectome, does not include the SEZ, where bitter-sensing axons arborize. Thus, we cannot use the connectome to analyze connectivity to second-order neurons. We added a sentence mentioning this (lines 285-287).

However, we have now used the connectome to trace downstream circuits from mlSEZt neurons, as described above and shown in Figure 9. These studies reveal that input from mlSEZt cells is first processed in the SLP and SIP before being relayed to other areas, including the SMP, lateral horn, mushroom body, and descending neurons. We identified hierarchical interconnections between mlSEZt cells, prominent feedback circuits, downstream neurons that are poised to integrate olfactory and taste input, and neurons controlling egg-laying that may integrate sweet and bitter input.

Discussion:It is surprising that the activation of the bitter projection neuron does not induce learning. What are the connections of these neurons in higher brain regions?

As described above, we have now conducted extensive studies to identify downstream circuits from mlSEZt neurons. Postsynaptic partners of these neurons are primarily confined to the SLP and SIP. Moving one synapse downstream, we do observe connections to the mushroom body (the learning and memory center), but not to PPL1 dopamine neurons that convey bitter reinforcement signals during learning. The absence of these connections may relate to the inability of mlSEZt to elicit learned aversion, and we have pointed this out in the text (lines 533-536).

Your data are of very high quality and very convincing. I am, however, not quite convinced by the interpretation.Looking at the data, I see that pharynx induces different behaviors (in particular learning) than the other subsets. Your trans-tango data provides a possible explanation. Your other data suggest that, indeed, leg and labellum drive very similar behavior (with perhaps the exception of 'post-light locomotion). This also fits to the trans-tango data overall.Therefore, it is difficult for me to follow your conclusion, e.g. abstract: 'These results suggest that different bitter inputs are selectively integrated early in the circuit, enabling the pooling of sensory information, and the circuit then diverges into multiple pathways that may have different behavioral roles.'

We apologize if our conclusions have not been presented clearly. In our view, the similarities between the three organs (they all affect proboscis extension, feeding, locomotion, and positional aversion) are more striking than the differences. However, we agree that the differences are important too, and we repeatedly mention them. Perhaps the level of similarity versus difference is a matter of interpretation (a glass half-full versus half-empty scenario), and by carefully laying out the results (e.g. the table in Figure 5A) the reader can form their own interpretation.

We believe that our conclusions are fully compatible with the reviewer’s interpretation. By saying that “different bitter inputs are selectively integrated early in the circuit”, we mean that these inputs are integrated within some pathways (e.g. all three organs project to the medial tract), but not every pathway must receive input from all organs. We refer to this as “selective integration”, and we have created a new diagram (Figure 5B) to visually depict this model. We have also added more explanations in the text to describe it. For example (lines 270-276):

“These results generally support an integrative model of taste processing (Figure 1B, left) in which bitter inputs from different organs converge onto common downstream pathways to drive a diverse set of aversive behaviors. However, the fact that not all neuronal subsets elicited all behaviors, or preferentially elicited some behaviors over others, suggests that some downstream pathways may receive selective or preferential input from specific bitter-sensing cells. We refer to this model as “selective integration” (Figure 5B), and it lies in between the two models shown in Figure 1B.”

The example that the reviewer mentions regarding the pharynx is the most clear case where a behavioral difference may be attributable to a difference in connectivity, and we have added a sentence to explicitly describe this point (lines 636-638).

Given the high quality of your data, I suggest condensing the manuscript and perhaps focussing on the differences between pharynx and the other neurons a little more. Moreover, using more inactivation experiments could be helpful, too, to distinguish the differences between these subsets in the different behaviors you study. Finally, the paper is divided into two parts, and I am not convinced that the sugar neuron data is needed for the second part and vice versa. It might make sense to split the manuscript into two.

In accordance with the reviewer’s suggestions, we have removed all of the sugar neuron data. We have also removed much of the basic characterization of bitter neuron activation (e.g. comparing starved versus fed) in order to focus on the role of different bitter neuron subsets.

In addition, we have added inactivation experiments where we silenced each subset of bitter neurons in a feeding assay (Figure 2E-G). We did not expect that this would have a strong effect because each subset alone is capable of suppressing feeding, suggesting that they act redundantly. Our results support this model: silencing bitter neuron subsets had relatively small and generally nonsignificant effects. We note that pharyngeal silencing showed the strongest trend toward increased feeding (lines 183-184), suggesting that there may be a stronger requirement for the pharynx than other organs.

We have not performed inactivation experiments with the other behaviors because this would require setting up new behavioral paradigms; our current setups only allow light stimulation. We would need to test natural bitter compounds and verify what kind of locomotor changes, preference, and learning they normally drive. We feel that setting up new assays is beyond the scope of this study. More importantly, our Gal4 lines do not allow us to silence 100% of bitter neurons in any given organ, and we do not anticipate observing strong effects by only partially silencing each organ (which is supported by our results in the feeding assay). This is the primary reason why we took the activation approach.

Reviewer #2:The authors tackle a particularly interesting question of how multiple taste cues from the same modality are integrated to form a behavioral response. The manuscript is scientifically sound, with well controlled experiments. The authors rely on optogenetic manipulations to define behavioral responses to different subsets of neurons. The field has tended to rely on relatively simple behavioral responses, such as PER, and a strength of this manuscript is the detailed characterization of locomotor behavior, learning, and reflexive feeding. This systematic characterization of bitter responsive neurons will be broadly useful to the field. In addition, the identification of second order neurons that selectively regulate PER provides a platform for studying integration. A number of concerns are noted that are readily addressable. First, there is a reliance on optogenetics that may not reflect the temporal or spatial patterns of natural compounds. The impact of the paper may be greater if these experiments were complemented with naturally occurring compounds. Second, as written much of the manuscript is descriptive, focusing on aspects that have previously been explored (such as the opposing valances of bitter and sugar neurons). Streamlining this part of the manuscript will make the significance more accessible. Overall, this manuscript provides a valuable resource for understanding how bitter tastants are coded, and will be of broad interest to groups researching sensory processing and behavioral regulation.

We thank the reviewer for these thoughtful comments. We have completely reworked the manuscript to streamline the sections that the reviewer felt are too descriptive and not sufficiently novel. First, we removed all data regarding the sugar neurons, as these results are not directly relevant to the rest of the paper. Second, we now present the activation of all bitter-sensing neurons as merely a control to analyze the effects of bitter neuron subsets, rather than presenting these results as novel findings. These results are no longer described in separate figures.

Regarding the optogenetic approach and testing natural compounds, please see our response to point #1 below.

1. The paper focuses on the complexity of taste coding, however, it does not firmly establish why the optogenetic approach is needed. There are many limitations to this approach, including the fact that the subsets of neurons, and their temporal activation may not reflect those induced by food. Additional justification of the use of this system is important.

We have now added some additional justification to the text (lines 121-128). We have several reasons for taking the activation approach, as opposed to performing neuronal silencing while flies taste natural bitter compounds:

1) Our goal was to test which behaviors can be driven by each subset of bitter neurons, rather than which subsets are required for behavior. The former question (addressed by neuronal activation) should reveal which neurons are connected to which behavioral circuits, even if different neurons have redundant functions. The latter question (addressed by neuronal silencing) will only reveal behavioral effects if neurons do not have redundant functions.

2) Related to the point above: Our Gal4 lines do not label 100% of the bitter neurons in any given organ. We would not expect to observe strong effects using neuronal silencing if there are still some functional bitter neurons in each organ, since redundant functions are very likely. Indeed, our new feeding experiments support this: we see very strong effects with neuronal activation and weak, nonsignificant effects with silencing.

3) The activation approach allows us to target neurons within any organ and precisely control the timing and duration of the stimulus. Conversely, presenting freely moving flies with natural bitter taste means that we cannot control which organs are stimulated, nor the stimulus timing/ duration. For example, if flies do not ingest the substrate then pharyngeal neurons will never be stimulated, and we cannot assess whether pharyngeal neurons are capable of driving behavior. How each organ samples bitter under natural conditions and which organs are required for each behavior are interesting questions, but ones that we plan to address in future studies.

We completely agree that optogenetic activation may not recapitulate natural activation, but this is a limitation of all optogenetic activation studies. For this reason, we first activated the entire population of bitter-sensing neurons and ensured that the behavioral effects are consistent with what we expect from previous studies of bitter taste. The concordance between natural and optogenetically evoked behaviors suggests that we are not activating bitter neurons in a totally unnatural way. We have also added new data using a different activation protocol: we compared the effect of continuous and 50 Hz pulsed light stimulation, and the behavioral effects were nearly identical (Figure 3 —figure supplement 1C-D and Figure 4 —figure supplement 1B). These results suggest that the behavioral effects we observe are not specific to a particular stimulation protocol.

Finally, we have complemented our activation approach with inactivation experiments for one type of behavior, feeding. The weak effects of neuronal silencing, as compared with the very strong effects of neuronal activation, support the idea that bitter neurons function redundantly.

2. The conclusions of Figure 2 are unclear. One possibility is that the activation of bitter neurons induces locomotion, which results in less time on those quadrants. The other is that they are actively avoiding the quadrant. Without addressing this, it is not clear what this adds to the conclusions beyond the data presented in Figure 1.

Figures 1 and 2 no longer exist in their previous form, so we are not sure if the reviewer would still have the same concern. In the revised manuscript, the effects of activating all bitter neurons are no longer presented as novel results; they are simply a control to test activation of bitter subsets.

We completely agree that positional aversion and locomotor stimulation could be linked, as we discussed in the original text. However, they are still very different behaviors displayed over different timescales, and it is possible that an active choice process contributes to positional aversion. It is not obvious that each bitter neuron subset should have identical effects in both assays, so we think it is reasonable to test their effects separately even if the results do end up being correlated. (Other behavioral measures we test are also intrinsically correlated, such as proboscis extension and feeding.) We have edited the text to include this rationale (lines 230-234).

3. The conclusion of the first four figures are factors that sweet and bitter neurons have oppositional valance, and that sugar is more potentially modulated by a huger state than bitter taste. These have been well described throughout the literature in multiple systems. At the authors' discretion, I suggest streamline these as it will make the novel aspects of the manuscript more accessible.

We have removed this entire section of the paper and incorporated the bitter neuron results into other figures. All sugar neuron results have been removed as well as comparisons between hunger states. The paper now begins with the activation of bitter neuron subsets.

4. Are the GAL4 lines used to characterize distinct subsets of neurons exclusively expressed in these body regions or preferentially expressed in them? I presume the former, but it would be valuable to reference this and/or show expression if it has not been thoroughly described across all regions.

The Gal4 lines have been thoroughly characterized in previous studies, and they are exclusively expressed in either the leg, labellum, or pharynx. We cited the references in the original text but we have now edited the text to be more explicit about the specificity of these lines (lines 135-139). We have also added our own images confirming the organ specificity of expression (Figure 1C).

5. It would be helpful to directly compare subsets of sweet and bitter second order neurons in figure 7 to gain insight into whether the circuit design is shared between the tastants.

We agree this would be interesting, but this has already been done by Snell et al. (*Curr Biol*, 2022).

Recommendations for the authors:1. In numerous cases the justification for the experimental approach is unclear. For example line 246 states 'We focused on bitter taste for these studies.' It would be helpful to know why the bitter taste was chosen over sweet.

We apologize for a lack of clarity, and we have tried to ensure that the justification for each experiment is clearly described. In the revised manuscript, we have rewritten the Introduction to justify why we are studying bitter taste. We first describe the known functional diversity of sweet and bitter neurons (within and across organs). We then describe the overall question (how different inputs within a modality are processed) and previous work on the sugar pathway, which supports the segregated processing model. Finally, we discuss how very few studies have addressed this question in the bitter pathway, creating a gap in knowledge that we aim to fill.

2. The introduction is very well written and frames the question of how tastants are detected. However, the focus primarily on the sweet and bitter modalities, and discusses how these are sensed by Grs. The complexity described includes the more recent descriptions of other modalities and new classes of gustatory receptors. It seems important to describe these as well in order to convey how the complexity of the system that is being dissected.

We have added a sentence mentioning other taste modalities (lines 56-59), but we did not want to add too much detail since our focus now is exclusively on the bitter taste pathway, and the introduction is already quite long.

3. Line 43 says 'like other sensory systems…' but the focus of the introduction is almost exclusively on taste. Much of this can be streamlined, and it would be helpful to add comparisons for context. For example, questions of sensory integration and combinatorial receptor activation have been addressed in much greater detail for olfaction and this may be worth mentioning.

The phrase “Like other sensory systems” was meant to be a transition from the broad first paragraph (discussing sensory systems in general) to focusing on taste specifically. We considered adding more examples from other senses to the Introduction, but the Introduction is already fairly long, and these examples seemed to distract from conveying the most relevant information. However, we have added a comparison to olfactory coding and other sensory modalities in the Discussion (lines 756-769).

4. Line 109: It's not clear what the first sentence means. Most tastes are likely to be feeding related. Is the point that most studies have used taste response to food components?

This sentence has been removed.

Reviewer #3:This manuscript by Deere et al. characterizes the behavior effects of optogenetic activation of sugar and bitter sensing GRNs. While activation of sugar-sensitive GRNs causes a brief decrease in locomotion, activation of bitter-sensitive GRNs increases locomotor activity in flies. Furthermore, the impact of sugar-GRN activation, but not bitter-GRN activation, is enhanced by food deprivation. The authors, also showed, pairing neutral odors with optogenetic activation of sugar-GRNs or bitter-GRNs can induce positive or negative valence to neutral odors in preference experiments in subsequent trials. After characterizing behavioral effects of optogenetic manipulation of sugar and bitter GRNs, the authors next focus on activation of bitter-sensing GRNs in different organs and discuss the differences and similarities between their behavioral effects. The authors observe some overlap between the behavior effects elicited by bitter-sensing GRN activation in different organs, legs, pharynx, labellum, etc. therefore they hypothesized bitter GRNs might share the same downstream neural circuits. Using transsynaptic labeling, they identified a group of neurons that receive input from different bitter sensing neurons, the mlSEZt neurons, and they showed that mlSEZt neurons showed ON and OFF responses to bitter tastant s but not sugar using calcium imaging. Interestingly, the response dynamics of mlSEZt neurons show different responses to bitter compounds, and the ON response of mlSEZt neurons habituated faster than the OFF response in response to repeated stimulation.Overall, this is a promising study, however, I have some major technical concerns about the way experiments have been conducted. Without having some clarification of GRN activity during optogenetic activation and anatomical characterization of the GRN drivers used in the study, it is hard to interpret what the results mean.

We thank the reviewer for their feedback. We believe that our responses below, in combination with new experiments and data, should fully address the reviewer’s concerns.

1. In all of the optogenetic activation experiments, the authors are using constant light stimulation. Previous papers (Inagaki 2014, Nature Methods) have shown that constant light stimulation causes suppression of action potentials in GRNs. Hence, it is not clear how the stimulus protocols used in the manuscript are impacting GRN activity. Authors need to show that either GRNs are actually activated and not shut down by the prolonged optogenetic stimulation protocols they used, via electrophysiology or calcium imaging. Alternatively, they can change their activation protocols by shortening the stimulus period or using pulses of light instead of constant light activation. Without having some evidence, the GRNs are actually active during the periods of optogenetic activation, it is hard to interpret the results of the experiments in Figure 1-5.

We agree that this is an important issue to consider. We first want to clarify that prolonged light stimulation is only a concern for the locomotion (5 sec stimulation) and innate aversion (30 sec stimulation) experiments. The learning experiments used 1 Hz light pulses, and the PER and feeding experiments used short light stimulations (1-1.5 sec). In addition, for locomotion assays we are primarily assessing changes at light onset (first 1 sec).

At the reviewer’s suggestion, we have now conducted experiments using 50 Hz pulsed light stimulation to activate all bitter-sensing neurons in the locomotor and choice assays (Figure 3 —figure supplement 1C-D and Figure 4 —figure supplement 1B). The behavioral effects of continuous versus pulsed activation were nearly identical, implying that continuous light activation does effectively activate the neurons. The same was true for sugar neurons, although these results are not included now that we are only focusing on bitter taste: either continuous or pulsed stimulation elicited the same behaviors.

In addition, it would be strange that we observed behavioral effects throughout the light stimulation period if the neurons were not actually activated. For 30 sec choice experiments, in no case does the level of aversion decrease during the stimulus period (Figure 4B-C), as would be expected if the neurons are shut down, and in most cases the aversion continuously increases.

We note that for the choice and learning experiments, we are emulating the exact protocols published by Aso and colleagues, whose results are highly regarded in the field (Aso et al., *eLife* 2014; Aso and Rubin, *eLife* 2016). Continuous 30 sec stimulation during choice assays also appears to be the protocol used to validate the effect of Chrimson in the original Chrimson paper (Klapoetke et al., 2014, Supp. Figure 16), as they do not mention anything about light pulses. In these papers, continuous 30 sec stimulation produced behavioral effects for a variety of neurons, including olfactory sensory neurons, olfactory projection neurons, bitter-sensing neurons, and mushroom body output neurons.

Finally, we note that the previous study mentioned by the reviewer (Inagaki et al., 2014) used a different optogenetic effector, ReachR. ReachR and Chrimson are derived from distinct opsins and are known to have different properties.

2. For sugar-GRN activations, it would be good to use another sugar-GRN, such as Gr5a to confirm Gr64f results.

All sugar neuron data have been removed from the paper, so this comment no longer applies.

3. In figure 5, the authors are activating subsets of bitter-GRNs and measuring the effects of optogenetic stimulations in various behaviors. I would like to see the expression patterns of GRN drivers used in these experiments in flies expressing Chrimson. Chrimson transgene used in these experiments has a Td-tomato tag, so authors can stain for this transgene and show us exactly which neurons they are stimulating. Without detailed anatomical characterization of the driver lines used in the experiments, it is again hard to interpret the results.

We completely agree that detailed anatomical characterization of the Gal4 lines is essential, and this characterization was performed by the previous studies we cited (Weiss et al., 2011; Ling et al., 2014; Chen and Dahanukar, 2017). They did the painstaking work of characterizing the presence or absence of expression in every labellar or tarsal sensilla and every cell type within the pharynx. We have also added our own images verifying the expression patterns of these lines, confirming that each line is expressed within only one of the three organs (Figure 1C).

4. See comment #2. The same concern here as Figure 5. Please show the expression pattern of the source GAL4s that are used for transsynaptic labeling. Without knowing the GRN expression, we can't interpret the results of downstream neurons.

We have added these images (Figure 6 —figure supplement 1). We observed the expected Gal4 expression patterns, with neurons in different organs showing different projection patterns that have been previously characterized (Kwon et al., 2014).

For further transparency, we have also added additional examples of *trans*-Tango staining for each line; this shows the consistency (and slight variability) between labeling in different brains.

5. In figure 8, are the ON and OFF responses of mlSEZt neurons displayed by arbors of the same neurons, or different neurons? I.e., is there any internal difference between different mlSEZt neurons, in terms of their response dynamics to bitter compounds?

We wish that we could compare the responses of different mlSEZt neurons, but this was not feasible because (1) the projections of different neurons are intermingled in the SEZ and SLP, and (2) the cell bodies did not show bitter responses. We think it is likely that the same cells generate both ON and OFF responses because this is the situation for bitter sensory neurons as well as downstream PPL1 dopaminergic neurons (Devineni et al., 2021).

Recommendations for the authors:1. In Figure 8 – —figure supplement 1C and D, it is better to include a diagram encircling the region of ROI in the brain. The arrows in Figure 8D are unclear.

We have added an image showing the ROIs (Figure 7 —figure supplement 1A).

2. For the experiments that involved optogenetic activation of both sugar sensing and bitter sensing neurons, how could you know that the behavior effect is not influenced by the possible different GAL4 expression levels of sugar sensing and bitter sensing GAL4 lines?

These experiments have been removed from the paper, so this comment no longer applies.

3. In Figure 1 – —figure supplement 1, there should be '(D)' in the last figure.

This figure has been reorganized, so this comment no longer applies.

4. In Figure 8, adding experiments with Gr66a-lexA > CsChrimson, mlSEZt-Split (the cleanest line) > TNT might be helpful in figuring out whether mlSEZt neurons are necessary for the locomotor responses, innate and learned preference. Also, as a negative control, Gr5a-lexA > CsChrimson, mlSEZt-Split > TNT can be tested if reagents are available.

As suggested, we have performed new experiments testing whether mlSEZt neurons are required for the effects of bitter neuron activation on locomotion, innate choice, and learning (Figure 8 —figure supplement 3). We did not expect that mlSEZt neurons would be required because their activation generally does not impact these behaviors. Indeed, none of the locomotor or preference behaviors were affected by mlSEZt silencing.

5. In Figure 8G, angular velocity significantly increases with high led intensity. Would it be possible to use higher led intensity to make sure that all mlSEZt populations are activated?

We have now performed these experiments (Figure 8 —figure supplement 2D). Using an even higher light intensity (44 µW/mm^2^) elicited an increase in angular velocity, similar to the previously tested “high intensity” (35 µW/mm^2^) stimulation. This result strengthens the evidence that mlSEZt neurons do exert a real effect on turning, and we have modified the text accordingly so that we are not downplaying this result. The higher intensity still did not affect forward velocity.

[Editors’ note: what follows is the authors’ response to the second round of review.]

Please address all reviewer comments with a detailed response and/or changes to the manuscript text and figures. If you have the data available for points 6 and 7 of Reviewer 2, please include them. Otherwise, please change the text to address the fact that this driver is not optimal.

We have addressed all of the reviewers’ comments; please see our detailed responses below. Our revisions include changes to the text and figures.

Points 6 and 7 from Reviewer 2 request that we perform functional imaging of mlSEZt cells using the cleaner split-Gal4 line, which we had already performed and is shown in Figure 7 —figure supplement 2C.

Reviewer #1 (Recommendations for the authors):The authors have done a commendable job in revising and thereby strongly improving the manuscript. As previously stated, the data is of high quality, and the results are convincing. The new results including the connectomics analysis are very interesting and integrate the findings in early sensory processing with higher-order processing. Moreover, the new optoPAD data including neuronal inhibition is helpful to make the point of functional redundancy and, in combination with the later results, highlights a difference in redundancy at different levels of behavioral performance and learning.The authors have addressed my previous concerns related to the first submission of this work in full. I am satisfied with their responses. In particular, I believe the paper in its current, more streamlined form is easier and more interesting to read after the removal of some of the data including the data on sugar sensing.

Thank you for these comments and for helping us improve the paper!

Reviewer #2 (Recommendations for the authors):My previous comments have been addressed. Here are some issues that still remain.Issues to be addressed in the revised version.1. The images are too small in Figure 1C. I suggest making three columns: One for the labellum only, one for the legs only, and one for the pharynx only. For the labellum column, authors can zoom in to the labellum, and for the pharynx, they can only zoom in to the pharynx. It will make the results in 1D clearer. As it is, neurons are not clearly visible.

We have modified the figure as suggested. We now show the labellum and pharynx separately, and images of all three organs have been cropped and enlarged to make the neurons more visible.

As an aside, we want to clarify that in Figure 1D, the classes and numbers of neurons labeled in each organ are based on previous studies that thoroughly characterized these Gal4 lines (Weiss et al., 2011; Ling et al., 2014; Chen and Dahanukar, 2017). This is stated in the figure legend. Our images confirm that each line is expressed in the correct organ-specific pattern and the cell numbers are similar, but we have not rigorously quantified the number and subtypes of labeled neurons.

2. Figure 2D, I don't see the point of plotting opto-ctrl and getting negative values; I think it is easier to plot ctrl-opto and plot positive values rather than negative values. Nothing will change, except the plots will look better, in my opinion. Right now, it is confusing to see negative values. Same argument for Figure 2G.

To us, the most important thing is to show the direction of the effect and to keep the convention consistent across figures. The effects of activating or silencing all bitter neurons go in opposite directions, so one of these two experiments is going to have a negative value regardless of whether we choose to plot opto-ctrl or ctrl-opto. It is true that more of the Gal4 lines show an effect with activation rather than silencing, hence more negative bars than positive bars, but either way the reader is going to need to interpret negative bars somewhere in the figure.

In choosing between opto-ctrl or ctrl-opto, we find opto-ctrl to be more intuitive since we consider the control to be the reference point, which should be subtracted. We understand that others may have a different opinion, but we believe the reader will be able to interpret the data either way.

Also, some of the negative controls seem to have a high variance.

Yes, we have learned that individual flies are highly variable in this assay! Even without optogenetic stimulation, flies choosing between two identical food sources often have a strong preference for one food or the other. Any optogenetic manipulations must override this innate variability in order to produce an effect. For each experiment we tested a minimum of 40 flies per genotype, which is a similar or higher number than other papers that used this assay.

Did the authors test for ANOVA and compare all controls and test animals with each other statistically? If yes, please indicate statistical significance in the plots.

Yes, for every feeding parameter the experimental flies were compared to both controls by one-way ANOVA. All significant effects are denoted with asterisks. We decided not to add “n.s.” to all nonsignificant bars to avoid cluttering up the figure, as was stated in the legend: “Experimental effects not labeled with an asterisk are not significant”. This is the same convention we used in the other behavior figures.

We have now edited the legend to be more explicit: “For each feeding measure, experimental flies were compared to both controls, and effects not labeled with an asterisk are not significant.” If the reviewer feels that “n.s.” needs to be added to each nonsignificant bar then we can add this, but we thought the lack of significance is implied by the lack of an asterisk (and stated in the legend).

3. I think Figure 6 is missing the VNC expression. Many leg neurons project to the VNC, showing only the brain Trans-tango signal is misleading.

Although certain types of tarsal gustatory neurons are known to arborize within the VNC (e.g. sugar-sensing and pheromone-sensing cells), previous studies suggest that tarsal bitter-sensing neurons pass through the VNC without making synaptic arborizations there (Kwon et al., 2014). For this reason, we chose to focus on second-order pathways in the brain and did not stain the VNC for trans-Tango expression. Nevertheless, we agree that this is a limitation that we should acknowledge explicitly, and we have added two sentences to that effect (lines 291-294).

4. I don't think there is any reason for 9 figures. Figures 5-6 can be combined or shown as supplementary figures.

We see the reviewer’s point, and initially we also were reluctant to include 9 figures. However, we believe that the current organization works best for conveying the story by ensuring that each figure contains a conceptually linked set of data. We agree that Figures 5 and 6 are small enough to be combined, but they are conceptually quite different and it might be confusing to combine them. Keeping them separate should not present an extra burden to the reader since we are presenting the same number of panels either way.

We know that the majority of readers do not look at the supplemental figures, so we are reluctant to put anything critical in the supplement. Even if it is just a summary/model like Figure 5, it is important to understanding the story.

5. I think it is too strong to say mlSEZt neurons are second-order bitter neurons. There is no experiment showing direct synaptic connectivity to bitter neurons. They can say bitter downstream neurons or show synaptic connectivity via EM or GRASP.

We have modified the text to address this concern. In the Results section where we first characterize mlSEZt, we now state (lines 383-388):

“Together, these results strongly suggest that the mlSEZt neurons labeled by *R29F12-Gal4* are second-order bitter neurons: they have an identical morphology to second-order bitter neurons labeled by *trans*-Tango, their projections overlap with the axon terminals of bitter-sensing neurons, and they show clear bitter responses. We will therefore refer to mlSEZt neurons as second-order bitter neurons, although it is formally possible that they are downstream bitter neurons that are not directly connected to bitter-sensing cells.”

In the Abstract and earlier parts of the text that readers will encounter before getting to this section, we have modified the wording to refer to mlSEZt as a “downstream” or “putative second-order” bitter neuron. We hope that these modifications, taken together, ensure that the language reflects the strength of the evidence.

6. Authors use a 'dirty' GAL4 for functional imaging experiments rather than the 'clean' split line. I think it might be better to use the cleaner split-GAL4 for functional imaging since many neurons are labeled in SEZ and hard to know which neurons' projections are recorded during imaging.

Although we initially performed functional imaging using the less specific R29F12 line, we also performed imaging using the cleaner split-Gal4 line. This experiment is shown in Figure 7 —figure supplement 2C and mentioned in the text (lines 408-410). The neurons labeled in the clean line respond to bitter taste and show the same dynamics as the less specific line.

In addition, we are confident that we are imaging the correct neurons even in the “dirty” R29F12 line. Even though the expression pattern may look dirty when compressed into a maximum intensity projection, when examining expression in all three dimensions it is clear that the mlSEZt projections in the relevant SEZ or SLP regions do not overlap with other neurons. Our functional imaging methods allow for high spatial resolution in all three dimensions (using two-photon imaging with a 60X objective). We are able to unequivocally identify mlSEZt cells based on their morphology and trace their projections into the SEZ or SLP. To convey this to the reader, we have added back a sentence that was removed in the previous version (lines 338-339): “The projections of mlSEZt neurons in the SEZ and SLP could be clearly identified based on their location and morphology.”

7. Effects on behavior upon silencing or activating the clean split are much weaker. I think functional imaging is essential to demonstrate these are the real downstream bitter neurons.

As mentioned above, we have imaged the mlSEZt neurons in the clean split-Gal4 line and shown that they respond to bitter taste (Figure 7 —figure supplement 2C). We agree that it is critical to know that these neurons are bitter-responsive.

8. Figure 9 is interesting, but it is not super relevant without really demonstrating whether mlSEZt neurons respond to bitter compounds. Also hard to demonstrate, the putative mLSEZt neurons they found in EM data are really mLSEZt neurons.

As mentioned above, we have shown that the mlSEZt neurons in the clean split-Gal4 line do respond to bitter taste (Figure 7 —figure supplement 2C).

Regarding the connectome data, we used both automated and manual searches to identify every neuron with even a vaguely similar morphology to mlSEZt cells, and after manually sorting through hundreds of cells, we are fairly confident that we have identified all neurons that have the correct morphology. Thus, the population of 21 neurons we identified should include the bitter-responsive mlSEZt cells. Moreover, the third-order neurons that we identified using the connectome (Figure 9H) closely resemble the postsynaptic neurons to mlSEZt labeled by trans-Tango (Figure 9B-C), suggesting that we have identified the correct mlSEZt neurons.

However, it is possible that not all of the 21 cells we identified represent bitter-responsive mlSEZt cells, as we mention in the text (lines 501-502). This is a caveat that needs to be acknowledged, but we don’t think it undermines our overall results given that the population of third-order neurons closely matches the trans-Tango staining, and individual downstream neurons would need to be functionally validated in future studies anyway. In addition, our finding that the 21 putative mlSEZt cells are highly interconnected suggests that most of the downstream neurons we identified are likely to be at least indirectly influenced by true mlSEZt cells.

Reviewer #3 (Recommendations for the authors):This revised manuscript has been significantly reorganized to focus on bitter taste. The streamlining of results and inclusion of clarifying data significantly strengthen this manuscript, and further our understanding of bitter taste processing. With these changes, the authors have addressed all of my previous concerns.

We appreciate this feedback and thank the reviewer for helping us improve the paper.